**Cite this article:** Álvarez-Armada N, Cameron CB, Bauer JE, Rahman IA. 2022 Heterochrony and parallel evolution of echinoderm, hemichordate and cephalochordate internal bars. *Proc. R. Soc. B* **289**: 20220258.

evolution, palaeontology

deuterostomes, Stylophora, pharyngeal openings, gill bars, homology, respiratory structures

**Author for correspondence:**
Nidia Álvarez-Armada
e-mail: nidia.alvarez.armada@gmail.com

# Heterochrony and parallel evolution of echinoderm, hemichordate and cephalochordate internal bars

Nidia Álvarez-Armada[1], Christopher B. Cameron[2], Jennifer E. Bauer[3] and Imran A. Rahman[4,5]

[1]School of Earth Sciences, University of Bristol, Bristol BS8 1QU, UK
[2]Département de sciences biologiques, Université de Montréal C.P. 6128, Succursale Centre-ville, Montréal, QC, Canada H3C 3J7
[3]University of Michigan Museum of Paleontology, Ann Arbor, MI 48109-1085, USA
[4]The Natural History Museum, London SW7 5BD, UK
[5]Oxford University Museum of Natural History, Oxford OX1 3PW, UK

NÁ-A, 0000-0001-8299-6355; CBC, 0000-0002-9810-7476; JEB, 0000-0002-6337-6270; IAR, 0000-0001-6598-6534

Deuterostomes comprise three phyla with radically different body plans. Phylogenetic bracketing of the living deuterostome clades suggests the latest common ancestor of echinoderms, hemichordates and chordates was a bilaterally symmetrical worm with pharyngeal openings, with these characters lost in echinoderms. Early fossil echinoderms with pharyngeal openings have been described, but their interpretation is highly controversial. Here, we critically evaluate the evidence for pharyngeal structures (gill bars) in the extinct stylophoran echinoderms *Lagynocystis pyramidalis* and *Jaekelocarpus oklahomensis* using virtual models based on high-resolution X-ray tomography scans of three-dimensionally preserved fossil specimens. Multivariate analyses of the size, spacing and arrangement of the internal bars in these fossils indicate they are substantially more similar to gill bars in modern enteropneust hemichordates and cephalochordates than to other internal bar-like structures in fossil blastozoan echinoderms. The close similarity between the internal bars of the stylophorans *L. pyramidalis* and *J. oklahomensis* and the gill bars of extant chordates and hemichordates is strong evidence for their homology. Differences between these internal bars and bar-like elements of the respiratory systems in blastozoans suggest these structures might have arisen through parallel evolution across deuterostomes, perhaps underpinned by a common developmental genetic mechanism.

## 1. Background

Elucidating the early evolution of deuterostomes is crucial for understanding the origins of the group to which we (vertebrates) belong, but has long proved challenging owing to the scarcity of unambiguous synapomorphies shared by all members of this major animal superphylum. Pharyngeal openings, which are outlets of the pharynx, are the only morphological character widely accepted as a deuterostome synapomorphy (e.g. [1–5] but see also [6]). These ciliated perforations in the pharyngeal wall take the form of either simple pores or dorsoventrally elongated slits (typically with specialized skeletal support) [2,7]. Among extant deuterostomes, pharyngeal openings are present in chordates and hemichordates, where they play an important role in feeding or respiration [8,9], but not in echinoderms [10]. Genetic evidence strongly supports the homology of deuterostome pharyngeal openings. Several transcription factors, including Pax1/9, Eya, FoxI, FoxC and FoxL1, are expressed during the development of the pharynx and gill pouches in chordates and hemichordates [11–14]. Hemichordates and chordates share the same pharyngeal gene cluster,

including four genes encoding transcription factors; the same cluster is conserved in the genomes of some echinoderms including asteroids, echinoids and holothuroids [15,16]. The latest common ancestor of deuterostomes is therefore hypothesized to have possessed pharyngeal openings, which were lost along the branch leading to extant echinoderms [1,10].

Putative pharyngeal openings have been described in various fossil deuterostomes (e.g. [4,17]), including some early echinoderms [18–20]. These fossil forms have the potential to inform on the sequence of acquisition of key deuterostome characters, but their interpretation remains controversial. Of particular significance are an extinct Palaeozoic group called the stylophorans. This clade comprises two traditional groupings, cornutes and mitrates, which are characterized by a plated calcite skeleton, an asymmetrical body and a single major appendage [18,19,21,22]. Although their phylogenetic position was historically contentious, with the group interpreted as echinoderms [20–22] or chordates [18,19], the presence of extensions of an echinoderm-type water vascular system and associated ambulacral structures in the proximal region of the appendage [23] unequivocally places stylophorans within Echinodermata. Cornute stylophorans also possess serially aligned body openings, but it is debated whether these are pharyngeal gill slits [24] or sutural pores like those found in other fossil echinoderms [22]. Two mitrate stylophorans, *Jaekelocarpus oklahomensis* and *Lagynocystis pyramidalis*, exhibit internal structures that have been interpreted as gill bars [18,19], comparable to those present in the pharynges of extant cephalochordates and enteropneust hemichordates [8]. These internal bars have alternatively been suggested to represent specialized respiratory and/or feeding structures [21,25,26] with no close analogues among other deuterostomes. These would be similar to the bar-like elements of the respiratory system of some extinct blastozoan echinoderms, such as blastoids and rhombiferans [27–29], which are generally not regarded as homologous [30].

Ontogenetic studies of modern deuterostomes reveal that pharyngeal bars first appear as simple pores in early developmental stages and are subsequently added posteriorly as growth continues [31]. The pharyngeal bars extend downwards from the dorsal side, with the associated pores elongated into slits (figure 1*j*) [32]. In blastoids, the bar-like structures of the respiratory system (hydrospires folds) are also added during ontogeny [33,34], with new folds added along the radiodeltoid suture and the depth of individual folds varying usually with the greatest depth at the newest folds (figure 1*e*) [33]. In rhombiferans, the bar-like structures of the respiratory system (rhombs) seem to also be added through ontogeny, with the oldest rhombs being the largest in size [35]. The number of pores increases during growth, however, the spacing between pores remains constant [35]. Unfortunately, there is no information available on the ontogeny of the internal bars in stylophorans, meaning it is not possible to use developmental data to test between alternative interpretations of these structures.

To evaluate the evidence for gill bars in stylophorans, we use X-ray tomography to measure and describe the morphology of the internal bars in the stylophorans *L. pyramidalis* and *J. oklahomensis*. For comparison, we also examine gill bars in extant enteropneust hemichordates and a cephalochordate, as well as morphologically similar bar-like elements of the respiratory systems in three fossil blastozoan echinoderms. We use linear discriminant analysis (LDA), principal component analysis (PCA) and pairwise analysis of variance (ANOVA) to quantify the similarity between these structures. The results provide new insights into stylophoran palaeobiology, with important implications for the appearance, function and evolution of internal bar-like structures in extinct echinoderms.

## 2. Methods

### (a) Samples

Three specimens of the Middle Ordovician stylophoran *L. pyramidalis* (NHMUK E29453, NHMUK E16107 and NHMUK E29043) were obtained on loan from the Natural History Museum, London (NHMUK). In addition, the Pennsylvanian stylophoran *J. oklahomensis* (UWBM 74305) was incorporated into our analysis using existing data available on DigiMorph (http://digimorph.org/specimens/Jaeckelocarpus_oklahomensis/). Complete adult specimens of the extant enteropneust hemichordates *Balanoglossus* sp. (low tide, Penrose Point State Park, Washington, August 2013) and *Schizocardium* sp. (subtidal, Corpus Christi Bay, Texas, March 2013) and the cephalochordate *Branchiostoma floridae* (shallow subtidal, Tampa Bay, Florida, June 1997) were collected and preserved in 70% ethanol. These samples were stained with phosphotungstic acid for 17 days.

Additionally, specimens of three Devonian blastozoan echinoderms, the fissiculate and spiraculate blastoids *Cryptoschisma* sp. (MGM-3383D) and *Hyperoblastus reimanni* (CMC IP 37404) and the Late Devonian rhombiferan *Strobilocystites polleyi* (CMC IP 36209), were selected for inclusion in our study. Specimens were obtained on loan from the Museo Geominero (MGM) and the Cincinnati Museum Center (CMC).

The preservational characteristics of the fossil samples are discussed in the electronic supplementary material, information.

### (b) X-ray tomography

Specimens of *L. pyramidalis*, *Balanoglossus* sp., *Schizocardium* sp., *B. floridae* and *S. polleyi* were imaged with X-ray micro-tomography using the Nikon Metrology HMX ST 225 micro-CT scanner at the Natural History Museum, London. *Jaekelocarpus oklahomensis* was scanned at the University of Texas High-Resolution X-Ray Computed Tomography Facility in 2001 (see [19] for methodological details). *Cryptoschisma* sp. and *H. reimanni* were imaged with synchrotron tomography using the TOMCAT beamline of the Swiss Light Source, Paul Scherrer Institut, Villigen, Switzerland. Details of scan settings are provided in electronic supplementary material, table S1.

### (c) Three-dimensional reconstructions

Tomographic datasets were used to digitally reconstruct the anatomy of 10 specimens using the SPIERS software suite [36]. This involved creating three-dimensional virtual models of the targeted internal structures for each specimen in SPIERSedit. The dimensions of all these structures were then measured in SPIERSview. See electronic supplementary material, information and figure S1 for details. Measurements of length were standardized against the total length of the pharynx or internal thecal cavity for each model, whereas measurements of width, depth and spacing were standardized against the diameter of the pharynx or internal thecal cavity.

### (d) Data analysis

Standardized measurements of the structures of interest were analysed quantitatively using R v. 3.6.3 [37]: PCA and LDA. The analyses were conducted using the stats (v. 3.6.3; [37]) and MASS (v. 7.3.51.5; [38]) packages. LDA results were further

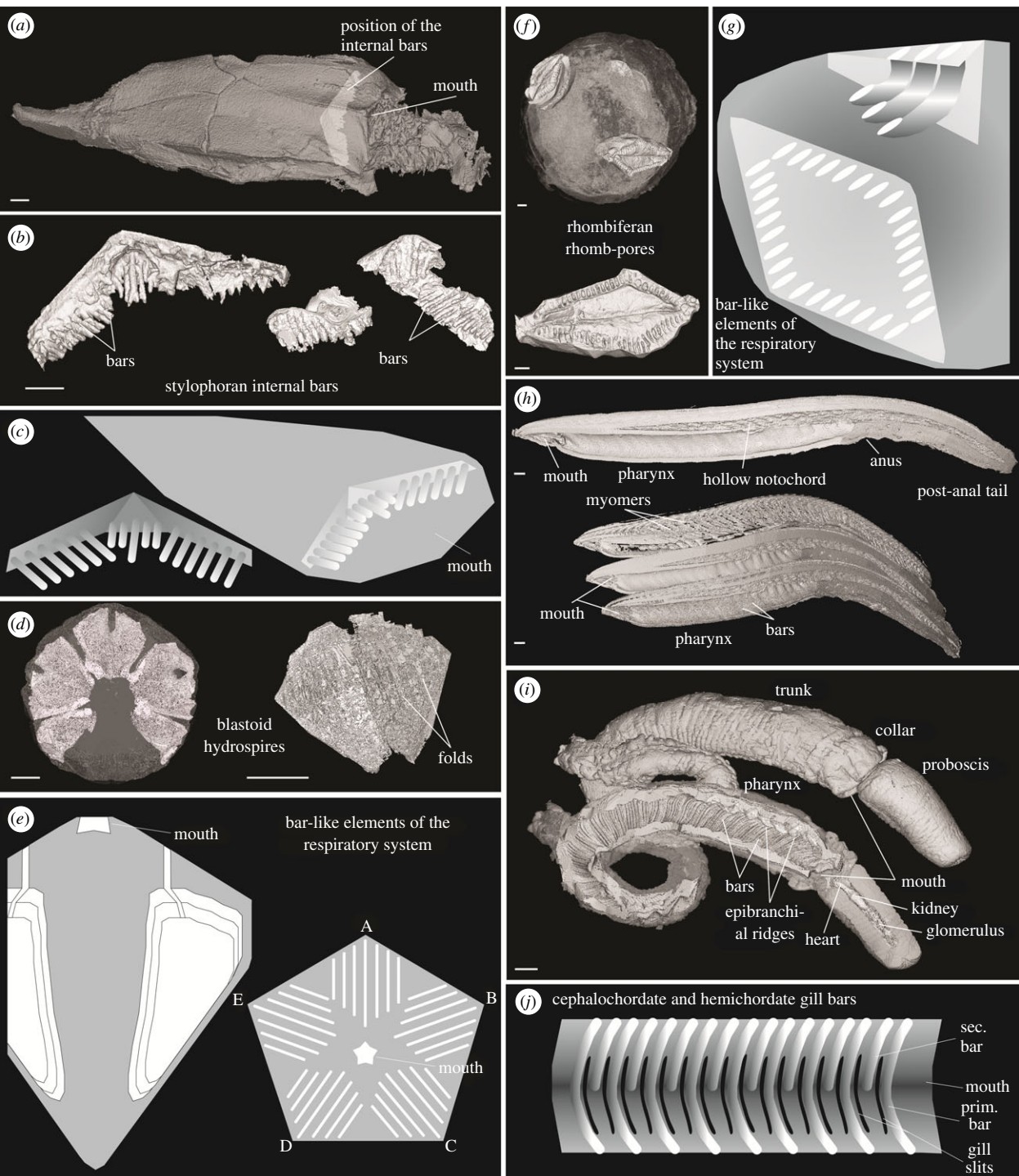

**Figure 1.** Virtual reconstructions of internal bars and bar-like structures in modern and fossil deuterostomes. (*a*) *Lagynocystis pyramidalis* (NHMUK E29043), external view showing the position of the internal bars. (*b*) Close-up of the internal bars in *Lagynocystis pyramidalis* (NHMUK E16107, left; NHMUK E29453, right). (*c*) Generalized diagram visualizing the distribution and morphological characteristics of the internal bars in *Lagynocystis pyramidalis*. (*d*) *Cryptoschisma* sp. (MGM-3383D), external view (left) and close-up of a single hydrospire group (right). (*e*) Generalized diagram visualizing the distribution and morphological characteristics of the hydrospires folds (bar-like structures) in *Hyperoblastus reimanni* and *Crysptoschisma* sp.; however, the latter lacks slits in the C-D interray. (*f*) *Strobilocystites polleyi* (CMC IP 36209), external view showing the position of the rhomb-pores (top) and close-up of a single rhomb-pore bearing complex (bottom). (*g*) Generalized diagram visualizing the distribution and morphological characteristics of the pores (bar-like structures) in *Strobilocystites polleyi*. (*h*) *Schizocardium* sp., lateral views showing external and internal morphological features. (*i*) *Branchiostoma floridae*, lateral views showing external and internal morphological features. (*j*) Generalized diagram visualizing the distribution and morphological characteristics of the gill bars in hemichordates and cephalochordates. Scale bars = 1 mm.

interrogated via partition plots for every combination of two variables using the klaR (v. 0.6.15; [39]) package. In addition, the normalized data were analysed using MANOVA, ANOVA at 95% confidence and *post hoc* testing (Holm–Bonferroni and Tukey 95% confidence tests) using the stats (v. 3.6.3; [37]) package. Further methodological details and R scripts are provided in electronic supplementary material, information.

## 3. Results

### (a) Anatomical description

The thecal cavities of the stylophorans house a series of repeating elongate bars, which are divided into three elliptical fields in *L. pyramidalis* (figure 1*a*–*c* and electronic

supplementary material, figure S2a,b) and two bilaterally symmetrical complexes in *J. oklahomensis* (electronic supplementary material, figure S2c). The pharynx walls of the hemichordates and cephalochordate are lined with pairs of parallel bars differentiated into primary and secondary bars that bifurcate ventrally (figure 1*h–j* and electronic supplementary material, figure S2g–i). The blastoids contain five pairs of hydrospires divided into a variable number of hollow folds, which connect to the exterior via slits or pores adjacent to the ambulacra (figure 1*d,e* and electronic supplementary material, figure S2d,e). The thecal cavity of the rhombiferan is connected to the exterior through rhomb-pore openings distributed within three complexes (figure 1*f,g* and electronic supplementary material, figure S2f). The major morphological characteristics of these structures are summarized in table 1. For detailed descriptions, refer to the anatomical descriptions in the electronic supplementary material, information and table S2.

## (b) Statistical analysis

The LDA plot shows that some taxa can be differentiated based on the dimensions of their internal bars or bar-like structures: the blastoids, cephalochordate and rhombiferan occupy distinct regions of the morphospace clearly separated from each other and other taxa, whereas the enteropneusts and stylophorans occupy a large region of the morphospace with a strong overlap (figure 2*a*). These trends are consistent with the PCA results (figure 2*b*). In LDA and PCA plots, depth and length are the main drivers of the disparity between groups (figure 2*c,d* and electronic supplementary material, figure S3). Squared cosines of PCA indicate that the importance of depth, width and spacing are high in the first dimension, while the second dimension is largely dominated by length (electronic supplementary material, information and figure S3). Boxplot and LDA partition plots further support these results for each grouping (electronic supplementary material, information and figures S4 and S5).

ANOVA indicates that there are no statistically significant differences in the width or length of the bars among the enteropneusts and stylophorans (electronic supplementary material, table S3). Conversely, the length of the folds in the blastoids and pores in the rhombiferan are significantly different from the length of the internal bars in all other groups (electronic supplementary material, table S3). The depth of bars in all the groupings shows statistically significant differences (electronic supplementary material, table S3). The spacing between bars in the stylophorans is not significantly different from the spacing of gill bars in the cephalochordates and enteropneusts (electronic supplementary material, table S3). Specimens of the same species (i.e. *L. pyramidalis*) show no statistically significant differences in any of the measurements analysed (electronic supplementary material, table S3).

## 4. Discussion

The results of our analyses demonstrate that the internal bars of the stylophorans *L. pyramidalis* and *J. oklahomensis* are morphologically very similar to the gill bars of modern cephalochordates and enteropneusts, supporting the hypothesis that these structures are homologous [18–20]. Multivariate analyses show a differentiated grouping in the morphospace formed by the cephalochordate, enteropneusts and stylophorans (figure 2). In particular, the internal bars in the cephalochordate, enteropneusts and stylophorans are similar in length, width and spacing (table 1 and electronic supplementary material, table S2). There is a clear correlation between the dimensions of the internal bars and the size of the pharynx or theca in stylophorans, enteropneusts and the cephalochordate (electronic supplementary material, information, figure S4 and table S2). By contrast, this trend is not observed in the blastoids and rhombiferan (electronic supplementary material, information and figure S4).

There are some notable differences between the internal bars of the stylophorans and the gill bars of extant deuterostomes. For instance, the secondary gill bars of cephalochordates and enteropneusts are not present in *L. pyramidalis* or *J. oklahomensis* (figure 1*b* and electronic supplementary material, figure S2a–c). There are also many fewer internal bars in the stylophorans (approx. 25 in *L. pyramidalis* and 8 in *J. oklahomensis*) than in the cephalochordate (approx. 250) and enteropneusts (approx. 130 in *Schizocardium* sp. and approx. 154 in *Balanoglossus* sp.). These differences may be the product of heterochrony because in the stylophorans the number of bars in adult animals remains much lower and the secondary bars do not appear to develop at any ontogenetic stage. In early developmental stages of enteropneusts the gills begin as a single pair of pores, with subsequent pairs added posteriorly during growth [31]. In cephalochordates the early ontogeny of gill slits is more complicated, with the earliest pores asymmetrical and arranged randomly; however, the number of pores still increases during ontogeny [32]. These pores are then extended into slits by the downward extension of the primary gill bars, resulting in structures that closely replicate the internal bars seen in the fossil stylophorans. In enteropneusts and cephalochordates, trunk coelomic diverticula (peripharyngeal coelomic diverticula) extend into the secondary gill bars, which are added as down growths between the primary gill bars [31]. Stylophorans lack secondary bars, which could also be due to a change in the relative timing of development of these structures, i.e. heterochrony or secondary loss [40]. The most significant difference between the bars of *L. pyramidalis* and *J. oklahomensis* versus those of hemichordates and cephalochordates is the composition. Stylophoran internal bars are thought to have originally composed of extracellular calcite [25,26]. In other stylophorans, the calcitic plates forming the skeleton develop the stereomic structure present in modern echinoderms [41], but there is no evidence for this in *L. pyramidalis* or *J. oklahomensis*. Pharyngeal bars in cephalochordates and enteropneusts are made from extracellular collagen [3,8,42]; however, there is no indication of soft tissues preserved in the fossils of *L. pyramidalis* and *J. oklahomensis*, subsequently if the internal bars had different composition (e.g. collagen) they would have not been preserved. Invertebrate collagen is not known to precede the development of mineralized tissue [43], but in vertebrate development, collagen precedes the mineralization of bone and in fish, the evolution of collagen gill arches precedes mineralized gill arches [44].

Multivariate analyses indicate that the gill bars and gill bar-like structures in the cephalochordate, enteropneusts and stylophorans show statistically significant differences from the bar-like elements of the respiratory systems in the blastoids and rhombiferan (figure 2 and electronic supplementary material, table S3). LDA and PCA cluster the blastoids and rhombiferan in separate groups (figure 2*a,b*). In addition, the blastoids show almost no overlap in the

**Table 1.** Anatomical description of internal bars and the bar-like structures in modern and fossil deuterostomes.

| taxon | no. bars | distribution | positioning | mean length (mm) | mean width (mm) | mean depth (mm) | mean spacing (mm) |
|---|---|---|---|---|---|---|---|
| Stylophora | | | | | | | |
| *Lagynocystis pyramidalis* | ≥25 | divided into three elliptical fields; five bars in the central field and at least 10 bars in each lateral | fields attach to the inner surface of different thecal plates | 0.711 | 0.121 | 0.144 | 0.086 |
| *Jaekelocarpus oklahomensis* | 8 | divided into two bilaterally symmetrical complexes of bars, each with four bars | projecting from the internal wall of adjacent thecal plates toward the interior of the theca | 0.564 | 0.099 | 0.303 | 0.104 |
| Hemichordata | | | | | | | |
| *Balanoglossus* sp. | ≥154 | extend from immediatly posterior of the collar to 1/3 of the trunk | wrapping both sides of the pharyngeal wall. Arranged in pairs | 2.458 | 0.068 | 0.093 | 0.093 |
| *Schizocardium* sp. | ≥130 | extend from immediatly posterior of the collar to 3/4 of the trunk | wrapping both sides of the pharyngeal wall. Arranged in pairs with shorter primary bars and longer secondary bars that bifurcate ventrally | 1.656 | 0.053 | 0.112 | 0.084 |
| Cephalochordata | | | | | | | |
| *Branchiostoma floridae* | ≥250 | extend from immediatly posterior to the mouth to 1/2 of the body | wrapping both sides of the pharyngeal wall. Arranged in pairs with shorter primary bars and longer secondary bars that bifurcate ventrally | 2.515 | 0.041 | 0.021 | 0.044 |
| Blastoidea | | | | | | | |
| *Cryptoschisma* sp. | ≥42 | distributed in eight hydrospire groups, usually arranged in groups of seven. Extend almost the entire length of the thecal cavity | ellipsoidal, multi-plated attached folds connected to the exterior through elongated slits adjacent to the ambulacra | 1.673 | 0.096 | 0.978 | 0.094 |
| *Hyperoblastus reimanni* | ≥18 | distributed in 10 hydrospire groups, arranged in groups of five. Extend almost the entire length of the thecal cavity | ellipsoidal, multi-plated attached folds connected to the exterior via sutural pores and elongated slits adjacent to the ambulacra | 2.004 | 0.200 | 0.538 | 0.281 |
| Rhombifera | | | | | | | |
| *Strobilocystites polleyi* | ≥120 | distributed among three rhomb-pore bearing complexes | 39 to 57 oval-shaped pores in each complex. Connect the interior to the exterior, lack evident connection to the mouth | 0.531 | 0.167 | 2.645 | 0.148 |

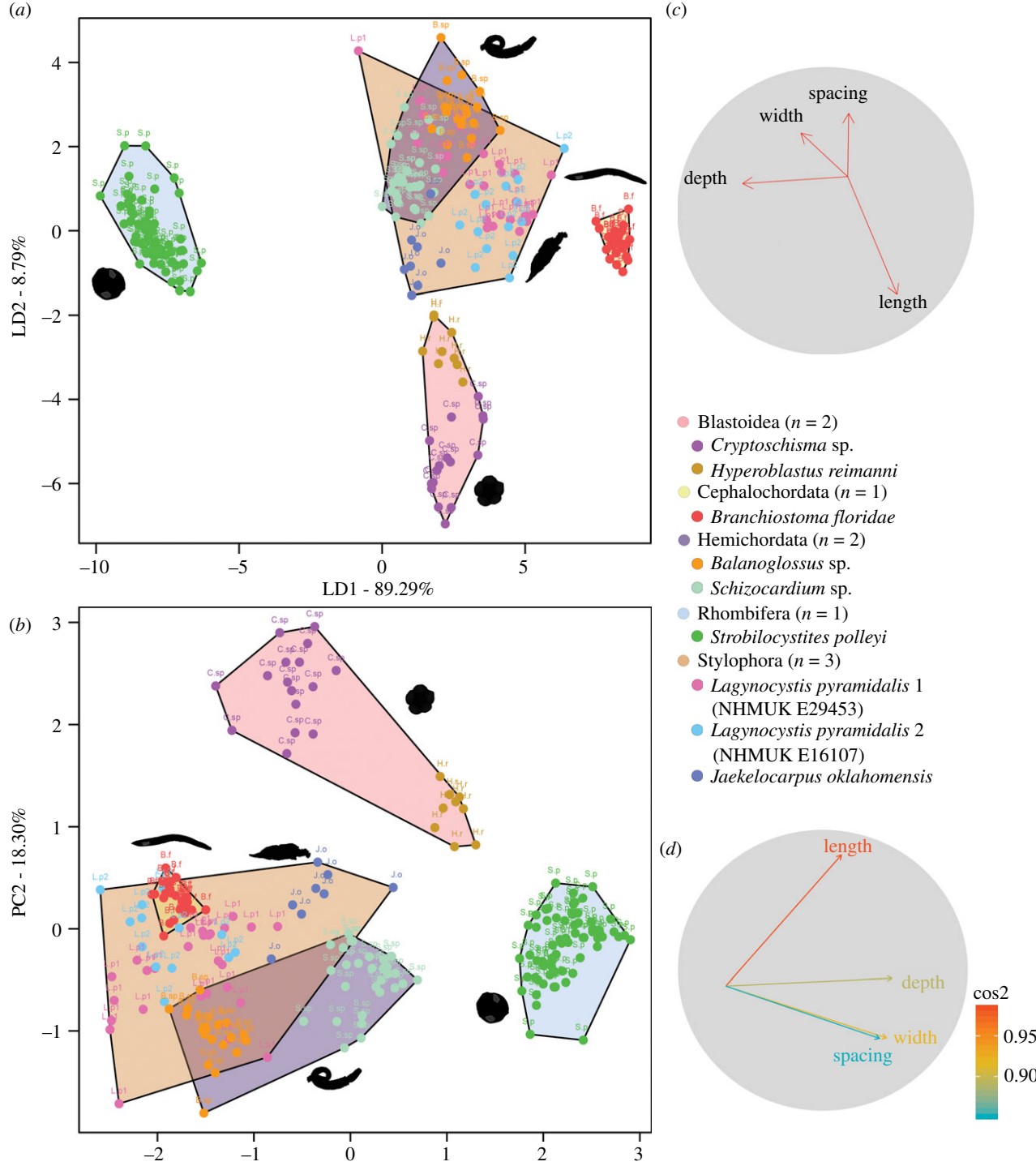

**Figure 2.** Linear discriminant analysis (LDA) (*a,c*) and principal component analysis (PCA) (*b,d*) of the dimensions (length, width, depth and spacing) of internal bars and bar-like structures in modern and fossil deuterostomes. (*a*) LDA plot of the resultant morphospace. (*b*) PCA plot of the resultant morphospace. (*c,d*) Biplot vectors showing the contribution of each dimension of the internal bars and bar-like structures to the variation in the dataset for LDA (*c*) and PCA with vectors colour coded by the square cosines contribution to the variation of the data (*d*). Abbreviations: B.f, *Branchiostoma floridae*; B.sp, *Balanoglossus* sp.; C.sp, *Cryptoschisma* sp.; H.r, *Hyperoblastus reimanni*; J.o, *Jaekelocarpus oklahomensis*; L.p1, *Lagynocystis pyramidalis* (NHMUK E29453); L.p2, *Lagynocystis pyramidalis* (NHMUK E16107); S.p, *Strobilocystites polleyi*.; S.sp, *Schizocardium* sp. (Online version in colour.)

morphospace (figure 2), particularly in the unconstrained PCA (figure 2*b*). This is most likely a result of the complex hydrospire morphology that is well documented as highly variable across taxa [28,29]. The average depth of the bar-like structures in the blastoids and rhombiferan is much larger than the enteropneusts, cephalochordate and stylophorans (table 1 and electronic supplementary material, table S2). The number of bar-like structures present in

blastoids is closer to that of the stylophorans than to the enteropneusts and cephalochordate (table 1). However, the bar-like elements of the respiratory system in blastoids are closely connected to the ambulacra and are arranged in groups with variable numbers of folds (figure 1*d,e* and electronic supplementary material, figure S2*d,e*), which differs from the arrangement of the internal bars in the stylophorans, enteropneusts and cephalochordate (figure 1*a–c,h–j* and

Deuterostomia

Chordata — Ambulacraria

Vertebrata  Urochordata  Cephalochordata  Pterobranchia  Enteropneusta  Stylophora  Blastozoa  crown Echinodermata

☆ primary and secondary bars. Slits. Many paired atria. Paired ectodermal pores. Synteny of pharynx and gill pore patterning genes.
☐ loss of primary and secondary bars. Single pair of atria. Single atrial and buccal siphons.
⬡ loss of primary and secondary gil bars in aminote vertebrates.

○ metapleural folds. Single pair of atria. One atrial pore.
◎ loss of primary and secondary bars. Single paired pores (*Cephalodiscus*).
╱ loss of secondary bars. Single atrium. Not recognizable exit pore.
✕ total loss of primary and secondary bars. Slits, atria and pores.

**Figure 3.** A generalized phylogenetic tree of the deuterostomes showing the evolution of key characters linked to the pharyngeal openings.

electronic supplementary material, figure S2a–c,g,h). There is little information available on the ontogeny of the internal bars in stylophorans, making direct comparisons with other echinoderms unfeasible; however, the structures preserved in adults greatly differ from blastozoans in their relative positioning, the nature of any associated external openings and overall morphology.

The function of the internal bars in the stylophorans is unclear. The gill bars of extant enteropneusts and cephalochordates are used in filter feeding [2,8,31,45]. In vertebrates, the tissues surrounding the gill bars are involved in ion exchange, respiration and excretion [46,47], with the first pairs of gill bars modified for talking, chewing and hearing [48]; they are also used for filter feeding in larval lampreys, herring and their relatives [8]. Thus, the function of gill bars varies across deuterostomes and is not constrained by their homology. Moreover, superficially similar parallel cylinders lined with rows of cilia, which are assumed to have arisen independently through parallel evolution, are also found in non-deuterostome metazoans and they can perform a range of different functions. The function of the bar-like structures of the blastozoans was primarily for respiration [29,49,50]. It was proposed that the internal bars of *J. oklahomensis* were lined with cilia, similar to gills in many modern animals, and they are close to large gaps in the theca that were interpreted as atrial openings [19,20], consistent with a role in filter feeding. However, the internal bars of *L. pyramidalis* are apparently not closely associated with any thecal openings (figure 1a and electronic supplementary material, figure S2a,b), suggesting a different function to filter feeding. The placement of the internal bars within the theca adjacent to the main appendage, which is assumed to have housed extensions of a water vascular system in life [22,23,25], points toward a possible role in feeding. Food particles captured by the appendage would have been conveyed into the thecal cavity through a mouth (inferred to have been situated at the base of the appendage), and the internal bars, which may have been covered in cilia [18,19], could have been involved in selecting particles and concentrating them into the gut (e.g. [21,22]).

The morphological characters shared by the Ordovician *L. pyramidalis* and Carboniferous *J. oklahomensis* are inferred to be derived within Stylophora. Earlier stylophorans, such as the Cambrian *Ceratocystis*, lack evidence of internal bars [51], and similar structures are not reported in any other groups of putative stem echinoderms [20,52], whereas fossils of *L. pyramidalis* and *J. oklahomensis* specimens are recurrently found with internal bars preserved. This may be because the

internal bars were composed of collagen in all other stylophorans (similar to the gill bars of extant deuterostomes), and hence would have had a much lower preservation potential than the calcitic bars of *L. pyramidalis* and *J. oklahomensis*. Collagenous gill bars are more decay resistant than other internal structures in extant hemichordates [53,54] and have been reported in some fossil forms [55–57], but these fossil specimens are known exclusively from Cambrian Burgess Shale-type Lagerstätten, from which stylophorans are entirely absent. Alternatively, the absence of internal bars in other stylophorans could reflect the secondary loss of these structures along the branches leading to most taxa. We suggest that differences between the internal gill bars in stylophorans and living deuterostomes, such as the number and morphology of the bars, could be the product of heterochronic evolution (figure 3). By contrast, we infer that bar-like elements of the respiratory systems in blastozoans, which are superficially similar to the internal bars of stylophorans and some extant deuterostomes, but statistically dissimilar in terms of their size and shape, could have evolved independently, perhaps by parallel evolution. Support for this parallel evolution, or deep-homology hypothesis comes from the conservation of pharyngeal gene clusters across deuterostomes, including several echinoderm groups [15,16]. The presence of pharyngeal development transcription factor gene synteny in living echinoderms that lack equivalent pharyngeal openings and gill bars strongly suggests that this gene arrangement was present in the common ancestor of deuterostomes, and raises the possibility that it could have been expressed in other echinoderm taxa. Thus, this highly conserved gene regulatory network might have been co-opted independently for the development of pharyngeal gill bar-like structures in blastozoan echinoderms.

These results bring us a step closer to resolving deep evolutionary traits within the deuterostome tree. Pharyngeal openings were present in the common ancestor of deuterostomes and secondarily lost in all echinoderms sometime after the Carboniferous. This has important implications for the sequence of acquisition of the fundamental echinoderm characters, with pharyngeal openings lost after the acquisition of a functional echinoderm-type water vascular system.

Data accessibility. The dataset associated with this study is available from the Dryad Digital Repository: https://doi.org/10.5061/dryad.nzs7h44rp [58].
The data are provided in the electronic supplementary material [59].

Authors' contributions. N.Á.-A.: conceptualization, data curation, formal analysis, methodology, writing—original draft, writing—review and

editing; C.B.C.: conceptualization, resources, supervision, writing—review and editing; J.E.B.: resources, writing—review and editing; I.A.R.: conceptualization, methodology, project administration, resources, supervision, writing—review and editing.

All authors gave final approval for publication and agreed to be held accountable for the work performed therein.

Conflict of interest declaration. We declare we have no competing interests.

Funding. This research was partially funded by a Career Development Grant PA-CD202101 from the Palaeontological Association awarded to N.Á.-A.

Acknowledgements. We thank Tim Ewin, Brenda Hunda, Cameron Schwalbach and Samuel Zamora for providing access to fossil material, Dan Sykes for carrying out micro-CT scans and Alberto Astolfo and Pablo Villanueva Perez for assistance with synchrotron tomography. We acknowledge Digimorph and the Thomas Burke Memorial Washington State Museum for supplying the micro-CT scan of *Jaekelocarpus oklahomensis* and the Paul Scherrer Institut, Villigen, Switzerland for the provision of synchrotron radiation beamtime on the TOMCAT beamline at the Swiss Light Source. We also thank our reviewers for their valuable comments and for strengthening the manuscript with their critics.

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
