## [Peer Review File · Proceedings of the Royal Society B: Biological Sciences]

Review History

RSPB-2021-1323.R0 (Original submission)

Review form: Reviewer 1 (Samuel Zamora)

Recommendation

Accept with minor revision (please list in comments)

Scientific importance: Is the manuscript an original and important contribution to its field?

Excellent

General interest: Is the paper of sufficient general interest?

Excellent

Quality of the paper: Is the overall quality of the paper suitable?

Excellent

Is the length of the paper justified?

Yes

Should the paper be seen by a specialist statistical reviewer?

No

Do you have any concerns about statistical analyses in this paper? If so, please specify them explicitly in your report.

No

It is a condition of publication that authors make their supporting data, code and materials available - either as supplementary material or hosted in an external repository. Please rate, if applicable, the supporting data on the following criteria.

Is it accessible?

Yes

Is it clear?

Yes

Is it adequate?

Yes

Do you have any ethical concerns with this paper?

No

Comments to the Author

This paper analyzes the presence of gill slits in an enigmatic group of early echinoderms known as stylophorans. The authors compare these structures using state of the art techniques with other deuterostomes and also with respiratory-like structures from other echinoderms.

The paper is well written, well supported and it is a novel approach with important phylogenetic implications for the understanding of early deuterostome evolution. I think this paper is an important contribution for Proceedings B. Nevertheless, I have several points that should be considered by the authors to strength their hypothesis.

The presence of gill slits in early echinoderms (especially in stylophorans) has been long debated from almost a century. There are two competing hypothesis, one suggesting that openings in stylophorans are respiratory structures and other suggesting that they were pharyngeal openings. In this aspect the approach of comparing these structures with those seen in hemichordates and cephalochordates using CT-scan and morphometrics is really interesting. My main concern on the pore-like structures seen in stylophorans is that earliest members of the group lack or not preserve the bars seen in *Lagynocystis* for example. The authors tackle this problem in lines 252-258; but I think they should expand such discussion. Could be these structures non-mineralize in early echinoderms, being made of cartilage? Why the external morphology of these structures vary so much in different stylophoran groups? etc.

Lines 65-66 – The authors indicate that stylophorans comprise “two clades” but it is largely demonstrated that they don’t. As far as I know from recent phylogenies cornutes are paraphyletic respect to mitrates. They are two clear traditional groups but phylogenetically speaking they are not two clades.

Comparison with stylophoran openings and those seen in other echinoderms (like blastoids and rhombiferans) is really welcome. First, I would like to emphasize that different types of respiratory structures in echinoderms are likely being analogous rather than homologous. So, in the supposed case that stylophoran openings were respiratory structures, they must be independently acquired in such group. Otherwise, I find very interesting the morphometric analysis showing that openings in stylophorans, hemichordates and cephalochordates cluster together in the figure 2. Paul (1968, *PALAEONTOLOGY* vol. 11) reported a large study of respiratory structures in cystoids with measures from different taxa (table 1). I just wonder if the authors are able to include such data in the analysis to reinforce the differences between respiratory structures and gill slits.

Congratulations on such interesting work,
Samuel Zamora

Review form: Reviewer 2 (Andrew Gillis)

Recommendation

Reject – article is not of sufficient interest (we will consider a transfer to another journal)

Scientific importance: Is the manuscript an original and important contribution to its field?

Acceptable

General interest: Is the paper of sufficient general interest?

Marginal

Quality of the paper: Is the overall quality of the paper suitable?

Acceptable

Is the length of the paper justified?

No

Should the paper be seen by a specialist statistical reviewer?

No

Do you have any concerns about statistical analyses in this paper? If so, please specify them explicitly in your report.

No

It is a condition of publication that authors make their supporting data, code and materials available - either as supplementary material or hosted in an external repository. Please rate, if applicable, the supporting data on the following criteria.

Is it accessible?

Yes

Is it clear?

Yes

Is it adequate?

Yes

Do you have any ethical concerns with this paper?

No

Comments to the Author

The evolution of pharyngeal arches is of fundamental importance to the early evolution of deuterostome animals - indeed, as the authors point out, some form of pharyngeal pore (and, by extension, arches the sit in between these pores) is the only unambiguous synapomorphy of deuterostome animals. It is now well accepted that the pharyngeal pores of cephalochordates and hemichordates are homologous (at the level of endodermal pouch derivatives) with the gill slits of vertebrates. Current deuterostome phylogenies would therefore suggest that the absence of such pores in extant echinoderms is a product of evolutionary loss. Consistent with this scenario is the presence of putative pharyngeal bars in the stylophorans, a group of stem-echinoderms. In this manuscript, the authors use comparative analysis of gill bar size and spacing in two

stylophorans and two extant deuterostomes (a cephalochordate and a hemichordate) to test the homology of these structures. They find that the gill bars of stylophorans are more similar (in morphospace) to those of cephalochordates and hemichordates than to the internal bars of another group of fossil echinoderms (the blastozoans). They conclude that stylophoran gill bars are homologous to those of cephalochordates and hemichordates, and make various speculations about heterochrony and parallel evolution in gill bar evolution.

I think that there is a valuable contribution in this manuscript - this sort of quantitative analysis of stylophoran gill bars has, to my knowledge, not been performed previously. However, I think that this article will be much better suited to a more specialist journal, and will require some additional fleshing out of the content before further consideration.

1. I think that a more nuanced Introduction to the pharyngeal structures of deuterostomes is needed in a paper like this. Deuterostome pharyngeal bars and gill pores are an excellent example of the hierarchical nature of homology: the pores form from endodermal outpockets (which are homologous to those of vertebrates, and which share gene expression features with those of vertebrates - particularly in relation to the conserved synthetic block of "pharyngeal" transcription factors mentioned in the text). And by extension, the tissue in between neighbouring pores (the pharyngeal arch) is also homologous - though there are additional tissues inside the pharyngeal arches of vertebrates that are not found within the pharyngeal arches of cephalochordates and hemichordates. This is touched upon in the Discussion (though not with sufficient detail), and also appears in the final figure (though I would argue, again, with insufficient treatment in the text). I think that the introduction should probably contain at least some brief discussion of how these pharyngeal structures form in hemichordates, cephalochordates and vertebrates, and the level(s) at which these structures are or are not homologous.

2. Also, the blastozoans just appear at the end of the Introduction, and it isn't entirely clear why. What are these animals, and why do they make a relevant comparison with stylophorans or other deuterostomes? This might be obvious to an echinoderm palaeontologist, but I don't think that others would necessarily immediately know the relevance of this group.

3. I am not a palaeontologist, and I only know about stylophorans from reading a few papers. I found it extremely difficult to understand the anatomy of these pharyngeal structures (and also of the internal bars of blastozoans) from Figure 1 and the supplemental figure. I think that it would be extremely helpful for non-palaeontologist/non-stylophoran experts if the authors could provide anatomical schematics or drawing to accompany their CT scans. This would provide some important organismal context for the animals that are being discussed here.

4. Also, is it not important to consider deformation/taphonomy when comparing quantitative morphology of fossil and extant structures? Especially when these structures differ in the nature of their tissues (soft vs. mineralised), and when only a very small number of individuals are being assessed?

5. While the statistical analysis performed seems sound, I have some concerns that the main points of the paper (i.e. heterochrony and parallel evolution of pharyngeal bars) are more speculation and hypothesis - i.e. the authors speculate about these patterns of evolution, rather than providing an abundance of new evidence for them. I don't think that there is anything wrong with this necessarily, but I do think that this is one of the aspects of the manuscript that makes it more suitable for a specialist journal, and perhaps less suitable for a general science journal like Proc. B. There is a passing mention, for example, that the absence of secondary gill bars in stylophorans is a product of heterochrony - but isn't it possible that the resolution of the anatomy from these CT scans isn't enough to distinguish between primary and secondary gill bars (if the latter are actually present)? Or rather than heterochrony (which implies a shift in the relative timing of developmental events through evolution), couldn't stylophorans have just lost secondary gill bars? With heterochrony in the title, I was expect a more fulsome discussion

around this point, but I felt that this wasn't thoroughly treated in the text.

6. Again, more nuance need in the Discussion re: homology. How is it known that the stylophoran gill bars were composed of calcite? And discussion of hemichordate pharyngeal bars of hemichordates being "composed of extracellular cartilage" is imprecise - the bars do contain a collagenous extracellular matrix, which appears to be secreted by the pharyngeal endoderm. But this does not equate with cartilage - again, homology at the level of general expression of SoxE/Col gene regulation does not mean homology at the level of tissue. These issues need to be discussed more thoroughly.

7. "We suggest that differences between the internal gill bars in stylophorans and living deuterostomes, such as the number and morphology of the bars, are the product of heterochronic evolution." Why?

8. "In contrast, we infer that bar-like elements of the respiratory systems in blastozoans, which are superficially similar to the internal bars of stylophorans and some extant deuterostomes, but statistically dissimilar in terms of their size and shape, evolved independently, perhaps by parallel evolution." Parallel evolution (i.e. independent evolution of morphologically similar structure, due to common underlying developmental mechanisms) of gill bars in blastozoans is untestable. So fine to speculate, but I don't think there is much to say about this other than "could be...".

Overall, I think that a more fulsome and nuanced analysis of pharyngeal bar morphology (integrating anatomy and morphometry), with discussions of homology, and possible scenarios of evolution could be a really nice contribution to the literature on this subject - but I think that it requires expansion of several of these points highlighted above, and is more suitable for a specialist journal.

Review form: Reviewer 3

Recommendation

Major revision is needed (please make suggestions in comments)

Scientific importance: Is the manuscript an original and important contribution to its field?

Excellent

General interest: Is the paper of sufficient general interest?

Excellent

Quality of the paper: Is the overall quality of the paper suitable?

Good

Is the length of the paper justified?

Yes

Should the paper be seen by a specialist statistical reviewer?

No

Do you have any concerns about statistical analyses in this paper? If so, please specify them explicitly in your report.

No

It is a condition of publication that authors make their supporting data, code and materials available - either as supplementary material or hosted in an external repository. Please rate, if applicable, the supporting data on the following criteria.

Is it accessible?

Yes

Is it clear?

Yes

Is it adequate?

Yes

Do you have any ethical concerns with this paper?

No

Comments to the Author

Please see attached PDF for comments.

Review form: Reviewer 4 (Thomas Stach)

Recommendation

Major revision is needed (please make suggestions in comments)

Scientific importance: Is the manuscript an original and important contribution to its field?

Acceptable

General interest: Is the paper of sufficient general interest?

Acceptable

Quality of the paper: Is the overall quality of the paper suitable?

Acceptable

Is the length of the paper justified?

Yes

Should the paper be seen by a specialist statistical reviewer?

Yes

Do you have any concerns about statistical analyses in this paper? If so, please specify them explicitly in your report.

Yes

It is a condition of publication that authors make their supporting data, code and materials available - either as supplementary material or hosted in an external repository. Please rate, if applicable, the supporting data on the following criteria.

Is it accessible?

No

Is it clear?

No

Is it adequate?

No

Do you have any ethical concerns with this paper?

No

Comments to the Author

I have very few specific comments to make, but some general ones.

Specifics:

The figures MUST be enlarged. Schizochardium in Table 1 should be Schizocardium. By the way, why are the species identity not known? At least for the extant species this should be remedied.

In the main text the references to the figures does not match the actual figure number.

Branchiostoma floridae in Table 1 should be italicized. In line 119: not the "Tomographic datasets" ... were ... reconstructed Rather, the tomographic datasets were used to digitally reconstruct the anatomy of the 10 specimens. Please rephrase.

Semi-specifics:

lines 199-201: during the early development ... begin as a single pair of pots, with subsequent pairs added posteriorly during growth. In Cephalochordates the ontogeny of gill slits is rather complex and starts out with asymmetric pores arranged so oddly, that evolutionary speculations abound. (I attach an old article of mine that has a drawing on page 10. There are much better articles to follow the development of gill slits, yet this is on my laptop now.)

line 236: not the bars are involve in the mentioned physiological processes, but the living tissues in the filaments are.

lines 252, 253: In my opinion (and I follow Jenner (2006; Unburdening evo-devo: ancestral attractions, model organisms, and basal baloney)), not animals are primitive/derived, but specific characters are. And that only in a certain context, i.e. comparatively.

General:

I found the morphological descriptions in the results section barren. For the fossils there is not enough detail given in terms of their position and relation to other identifiable structures within the fossil to gauge any similarity of the respective fossil structures to gill bars in extant taxa. In fact, at least in my opinion, the results section "Anatomical description" not only delivers very scarcely, what the title promises, but it also does not clearly distinguish between results and interpretation. In fossils, anatomical denominations, such as slits, pores, ambulacra, thecae cavity, hollow folds etc. are already interpretations, because all you have are differences and lines and shapes in a rock. Most often, even hollow spaces or cavities are now filled by rock. Thus, either justify the interpretations in the results section by plausible descriptions and supportive figures, or state clearly, that you follow the published interpretation of another researcher.

The discussion parts about heterochrony, parallel evolution, and deep homology do not convince me. There is neither enough consideration of morphological complexity/specifics to allow for the conclusion of parallel evolution. Nor is there any indication of ontogenetic shifts in the development of any of the structures implicitly proposed to be homologous to allow for the descriptor "heterochrony". A more detailed phylogenetic account would be necessary in that case as well. At least, I would expect to include (some) ascidians, cyclostomes, hagfishes, and gnathostomes in such speculations. I am also puzzled by the brazen interpretation of transcription factor genes as indicative of underlying parallel evolution of bar-like structures. Naturally, nothing is known about the context of these genes in blastozoan development. (As a remark on the side, the fact that these genes are present in echinoderms without any gill bar like structures, shows that these transcription factors can evolve independently from the structures. Consequently, similarity in gene expression would not (necessarily) indicate homology of the resulting structures.)

As an unashamed move of self-advertising, I also attach another article, a book chapter, that might be of interest, because it is on the general subject of morphological evolution in deuterostomes. And relatively obscure.

I do hope this is not too negative but worthy of consideration. If you find it too negative, please kindly ignore it.

Best wishes and sincere regards
Thomas Stach

Decision letter (RSPB-2021-1323.R0)

08-Nov-2021

Dear Dr Álvarez Armada:

I am writing to inform you that your manuscript RSPB-2021-1323 entitled "Heterochrony and parallel evolution of echinoderm, hemichordate and cephalochordate internal bars" has, in its current form, been rejected for publication in Proceedings B.

This action has been taken on the advice of referees, who have recommended that substantial revisions are necessary. With this in mind we would be happy to consider a resubmission, provided the comments of the referees are fully addressed. However please note that this is not a provisional acceptance.

Sincerely,
Dr Locke Rowe
<mailto:proceedingsb@royalsociety.org>

Associate Editor
Board Member: 1
Comments to Author:

Four experts in the field have reviewed your manuscript. While overall, they find the topic of the work interestingly, they have identified several weaknesses. These critiques concern key aspects of the paper, such as the evidence supporting the heterochrony and parallel evolution. Considering the reviewers' comments, I cannot recommend the manuscript for publication.

Reviewer(s)' Comments to Author:

Referee: 1

Comments to the Author(s)

This paper analyzes the presence of gill slits in an enigmatic group of early echinoderms known as stylophorans. The authors compare these structures using state of the art techniques with other deuterostomes and also with respiratory-like structures from other echinoderms.

The paper is well written, well supported and it is a novel approach with important phylogenetic implications for the understanding of early deuterostome evolution. I think this paper is an important contribution for Proceedings B. Nevertheless, I have several points that should be considered by the authors to strength their hypothesis.

The presence of gill slits in early echinoderms (especially in stylophorans) has been long debated from almost a century. There are two competing hypothesis, one suggesting that openings in stylophorans are respiratory structures and other suggesting that they were pharyngeal openings. In this aspect the approach of comparing these structures with those seen in hemichordates and cephalochordates using CT-scan and morphometrics is really interesting. My main concern on the pore-like structures seen in stylophorans is that earliest members of the group lack or not preserve the bars seen in *Lagynocystis* for example. The authors tackle this problem in lines 252-258; but I think they should expand such discussion. Could be these structures non-mineralize in early echinoderms, being made of cartilage? Why the external morphology of these structures vary so much in different stylophoran groups? etc.

Lines 65-66 – The authors indicate that stylophorans comprise “two clades” but it is largely demonstrated that they don't. As far as I know from recent phylogenies cornutes are paraphyletic respect to mitrates. They are two clear traditional groups but phylogenetically speaking they are not two clades.

Comparison with stylophoran openings and those seen in other echinoderms (like blastoids and rhombiferans) is really welcome. First, I would like to emphasize that different types of respiratory structures in echinoderms are likely being analogous rather than homologous. So, in the supposed case that stylophoran openings were respiratory structures, they must be independently acquired in such group. Otherwise, I find very interesting the morphometric analysis showing that openings in stylophorans, hemichordates and cephalochordates cluster together in the figure 2. Paul (1968, *PALAEONTOLOGY* vol. 11) reported a large study of respiratory structures in cystoids with measures from different taxa (table 1). I just wonder if the authors are able to include such data in the analysis to reinforce the differences between respiratory structures and gill slits.

Congratulations on such interesting work,
Samuel Zamora

Referee: 2

Comments to the Author(s)

The evolution of pharyngeal arches is of fundamental importance to the early evolution of deuterostome animals - indeed, as the authors point out, some form of pharyngeal pore (and, by extension, arches the sit in between these pores) is the only unambiguous synapomorphy of deuterostome animals. It is now well accepted that the pharyngeal pores of cephalochordates and hemichordates are homologous (at the level of endodermal pouch derivatives) with the gill slits of vertebrates. Current deuterostome phylogenies would therefore suggest that the absence of such pores in extant echinoderms is a product of evolutionary loss. Consistent with this scenario is the presence of putative pharyngeal bars in the stylophorans, a group of stem-echinoderms. In this manuscript, the authors use comparative analysis of gill bar size and spacing in two stylophorans and two extant deuterostomes (a cephalochordate and a hemichordate) to test the homology of these structures. They find that the gill bars of stylophorans are more similar (in

morphospace) to those of cephalochordates and hemichordates than to the internal bars of another group of fossil echinoderms (the blastozoans). They conclude that stylophoran gill bars are homologous to those of cephalochordates and hemichordates, and make various speculations about heterochrony and parallel evolution in gill bar evolution.

I think that there is a valuable contribution in this manuscript - this sort of quantitative analysis of stylophoran gill bars has, to my knowledge, not been performed previously. However, I think that this article will be much better suited to a more specialist journal, and will require some additional fleshing out of the content before further consideration.

1. I think that a more nuanced Introduction to the pharyngeal structures of deuterostomes is needed in a paper like this. Deuterostome pharyngeal bars and gill pores are an excellent example of the hierarchical nature of homology: the pores form from endodermal outpockets (which are homologous to those of vertebrates, and which share gene expression features with those of vertebrates - particularly in relation to the conserved synthetic block of "pharyngeal" transcription factors mentioned in the text). And by extension, the tissue in between neighbouring pores (the pharyngeal arch) is also homologous - though there are additional tissues inside the pharyngeal arches of vertebrates that are not found within the pharyngeal arches of cephalochordates and hemichordates. This is touched upon in the Discussion (though not with sufficient detail), and also appears in the final figure (though I would argue, again, with insufficient treatment in the text). I think that the introduction should probably contain at least some brief discussion of how these pharyngeal structures form in hemichordates, cephalochordates and vertebrates, and the level(s) at which these structures are or are not homologous.

2. Also, the blastozoans just appear at the end of the Introduction, and it isn't entirely clear why. What are these animals, and why do they make a relevant comparison with stylophorans or other deuterostomes? This might be obvious to an echinoderm palaeontologist, but I don't think that others would necessarily immediately know the relevance of this group.

3. I am not a palaeontologist, and I only know about stylophorans from reading a few papers. I found it extremely difficult to understand the anatomy of these pharyngeal structures (and also of the internal bars of blastozoans) from Figure 1 and the supplemental figure. I think that it would be extremely helpful for non-palaeontologist/non-stylophoran experts if the authors could provide anatomical schematics or drawing to accompany their CT scans. This would provide some important organismal context for the animals that are being discussed here.

4. Also, is it not important to consider deformation/taphonomy when comparing quantitative morphology of fossil and extant structures? Especially when these structures differ in the nature of their tissues (soft vs. mineralised), and when only a very small number of individuals are being assessed?

5. While the statistical analysis performed seems sound, I have some concerns that the main points of the paper (i.e. heterochrony and parallel evolution of pharyngeal bars) are more speculation and hypothesis - i.e. the authors speculate about these patterns of evolution, rather than providing an abundance of new evidence for them. I don't think that there is anything wrong with this necessarily, but I do think that this is one of the aspects of the manuscript that makes it more suitable for a specialist journal, and perhaps less suitable for a general science journal like Proc. B. There is a passing mention, for example, that the absence of secondary gill bars in stylophorans is a product of heterochrony - but isn't it possible that the resolution of the anatomy from these CT scans isn't enough to distinguish between primary and secondary gill bars (if the latter are actually present)? Or rather than heterochrony (which implies a shift in the relative timing of developmental events through evolution), couldn't stylophorans have just lost secondary gill bars? With heterochrony in the title, I was expect a more fulsome discussion around this point, but I felt that this wasn't thoroughly treated in the text.

6. Again, more nuance need in the Discussion re: homology. How is it known that the stylophoran gill bars were composed of calcite? And discussion of hemichordate pharyngeal bars of hemichordates being "composed of extracellular cartilage" is imprecise - the bars do contain a collagenous extracellular matrix, which appears to be secreted by the pharyngeal endoderm. But this does not equate with cartilage - again, homology at the level of general expression of SoxE/Col gene regulation does not mean homology at the level of tissue. These issues need to be discussed more thoroughly.

7. "We suggest that differences between the internal gill bars in stylophorans and living deuterostomes, such as the number and morphology of the bars, are the product of heterochronic evolution." Why?

8. "In contrast, we infer that bar-like elements of the respiratory systems in blastozoans, which are superficially similar to the internal bars of stylophorans and some extant deuterostomes, but statistically dissimilar in terms of their size and shape, evolved independently, perhaps by parallel evolution." Parallel evolution (i.e. independent evolution of morphologically similar structure, due to common underlying developmental mechanisms) of gill bars in blastozoans is untestable. So fine to speculate, but I don't think there is much to say about this other than "could be...".

Overall, I think that a more fulsome and nuanced analysis of pharyngeal bar morphology (integrating anatomy and morphometry), with discussions of homology, and possible scenarios of evolution could be a really nice contribution to the literature on this subject - but I think that it requires expansion of several of these points highlighted above, and is more suitable for a specialist journal.

Referee: 3

Comments to the Author(s)

Please see attached PDF for comments.

Referee: 4

Comments to the Author(s)

I have very few specific comments to make, but some general ones.

Specifics:

The figures MUST be enlarged. Schizochardium in Table 1 should be Schizocardium. By the way, why are the species identity not known? At least for the extant species this should be remedied. In the main text the references to the figures does not match the actual figure number. *Branchiostoma floridae* in Table 1 should be italicized. In line 119: not the "Tomographic datasets" ... were ... reconstructed Rather, the tomographic datasets were used to digitally reconstruct the anatomy of the 10 specimens. Please rephrase.

Semi-specifics:

lines 199-201: during the early development ... begin as a single pair of pots, with subsequent pairs added posteriorly during growth. In Cephalochordates the ontogeny of gill slits is rather complex and starts out with asymmetric pores arranged so oddly, that evolutionary speculations abound. (I attach an old article of mine that has a drawing on page 10. There are much better articles to follow the development of gill slits, yet this is on my laptop now.)

line 236: not the bars are involve in the mentioned physiological processes, but the living tissues in the filaments are.

lines 252, 253: In my opinion (and I follow Jenner (2006; Unburdening evo-devo: ancestral attractions, model organisms, and basal baloney)), not animals are primitive/derived, but specific characters are. And that only in a certain context, i.e. comparatively.

General:

I found the morphological descriptions in the results section barren. For the fossils there is not enough detail given in terms of their position and relation to other identifiable structures within the fossil to gauge any similarity of the respective fossil structures to gill bars in extant taxa. In fact, at least in my opinion, the results section "Anatomical description" not only delivers very scarcely, what the title promises, but it also does not clearly distinguish between results and interpretation. In fossils, anatomical denominations, such as slits, pores, ambulacra, thecae cavity, hollow folds etc. are already interpretations, because all you have are differences and lines and shapes in a rock. Most often, even hollow spaces or cavities are now filled by rock. Thus, either justify the interpretations in the results section by plausible descriptions and supportive figures, or state clearly, that you follow the published interpretation of another researcher.

The discussion parts about heterochrony, parallel evolution, and deep homology do not convince me. There is neither enough consideration of morphological complexity/specifics to allow for the conclusion of parallel evolution. Nor is there any indication of ontogenetic shifts in the development of any of the structures implicitly proposed to be homologous to allow for the descriptor "heterochrony". A more detailed phylogenetic account would be necessary in that case as well. At least, I would expect to include (some) ascidians, cyclostomes, hagfishes, and gnathostomes in such speculations. I am also puzzled by the brazen interpretation of transcription factor genes as indicative of underlying parallel evolution of bar-like structures. Naturally, nothing is known about the context of these genes in blastozoan development. (As a remark on the side, the fact that these genes are present in echinoderms without any gill bar like structures, shows that these transcription factors can evolve independently from the structures. Consequently, similarity in gene expression would not (necessarily) indicate homology of the resulting structures.)

As an unashamed move of self-advertising, I also attach another article, a book chapter, that might be of interest, because it is on the general subject of morphological evolution in deuterostomes. And relatively obscure.

I do hope this is not too negative but worthy of consideration. If you find it too negative, please kindly ignore it.

Best wishes and sincere regards

Thomas Stach

Author's Response to Decision Letter for (RSPB-2021-1323.R0)

See Appendix A.

RSPB-2022-0258.R0

Review form: Reviewer 1 (Samuel Zamora)

Recommendation

Accept as is

Scientific importance: Is the manuscript an original and important contribution to its field?

Excellent

General interest: Is the paper of sufficient general interest?

Excellent

Quality of the paper: Is the overall quality of the paper suitable?

Excellent

Is the length of the paper justified?

Yes

Should the paper be seen by a specialist statistical reviewer?

No

Do you have any concerns about statistical analyses in this paper? If so, please specify them explicitly in your report.

No

It is a condition of publication that authors make their supporting data, code and materials available - either as supplementary material or hosted in an external repository. Please rate, if applicable, the supporting data on the following criteria.

Is it accessible?

Yes

Is it clear?

Yes

Is it adequate?

Yes

Do you have any ethical concerns with this paper?

No

Comments to the Author

Dear authors,

I have read again the MS and response to referees, and you have addressed all my concerns in a satisfactory way. I believe this would be an important paper for Early Echinoderms Evolution with wider implications for the understanding of deuterostome origin. New morphological data is important and statistical approach is novel for the field. Congratulations!

Best wishes,

Samuel

Review form: Reviewer 5

Recommendation

Reject - article is scientifically unsound

Scientific importance: Is the manuscript an original and important contribution to its field?

Good

General interest: Is the paper of sufficient general interest?

Marginal

Quality of the paper: Is the overall quality of the paper suitable?

Marginal

Is the length of the paper justified?

No

Should the paper be seen by a specialist statistical reviewer?

No

Do you have any concerns about statistical analyses in this paper? If so, please specify them explicitly in your report.

No

It is a condition of publication that authors make their supporting data, code and materials available - either as supplementary material or hosted in an external repository. Please rate, if applicable, the supporting data on the following criteria.

Is it accessible?

Yes

Is it clear?

Yes

Is it adequate?

No

Do you have any ethical concerns with this paper?

No

Comments to the Author

This quantitative study of fossils of extinct stylophorans (*Lagynocystis* and *Jaekelocarpus*), widely accepted as deuterostomes, utilized sophisticated technology to image and reconstruct internal anatomical structures thought to be involved in suspension feeding or respiration. While these have been called gill bars by earlier workers. The actual placement of the group as echinoderms or early chordates is a matter still unsettled in the literature but is key to the overall conclusions. The disputed affinity is not mentioned. Nevertheless, the current study considers them to be early echinoderms and presents quantitative measurements of dimensions of these gill bars and compares them with gill bars of extant hemichordates and cephalochordates. Additional datasets generated from “bar-like elements” thought to be respiratory structures in 2 other definitive, but extinct, echinoderm groups (blastoids and rhombiferan) are added for additional comparison.

The images of reconstructions of the stylophoran *Lagynocystis* are exquisite (but too small) and add to earlier, similar images (Dominguez et al) of the other stylophoran (*Jaekelocarpus*) used in the dataset.

The multidimensional analyses show that dimensions of gill bars in stylophorans clearly and strongly overlap those of the chordate groups but not those of the echinoderm groups. These data support morphological homology of the gill bars of stylophorans and chordates. They could also be convergent structures.

Based on this dataset and the assumption that stylophorans are echinoderms the manuscript argues that “the latest common ancestor of echinoderms, hemichordates and chordates was a bilaterally symmetrical worm with pharyngeal openings, with these characters lost in echinoderms” (MS Abstract, lines 23-25). The argument is complex and hard to follow as presented. Several current developmental phenomena/processes (heterochrony and deep homology) are invoked to rationalize the inference that gill bars should be present in the last common ancestor of chordates and echinoderms.

I am a reviewer of what appears to be a thoroughly reviewed and revised MS; comments from 4 other reviewers and responses of authors were included in my “review package”. While I’ve

read these comments and responses I do not have the time or expertise to arbitrate the revisions. I will give a few comments on my reaction to the current MS.

The images and datasets are important and valuable contributions. The images in the supplementary material are also important for context. Perhaps they should be combined with those of the main text (and those in Dominquez et al) and made into a larger manuscript. (I think a more expansive manuscript directed to a more specialized audience would have a greater, longer lasting impact (than the present MS) on thinking about gill bars and how stylophorans may have worked).

The phylogenetic position of stylophorans is a key assumption and not sufficiently discussed in the MS. So the whole argument is "If stylophorans are echinoderms then" What if they are early chordates? Related is that stylophorans studied here are not the earliest of their clade and the gill bars described here are not reported from earlier group members. That's curious and not really addressed.

"Heterochrony" is defined as a change in timing of expression of a trait relative to that in a related lineage. I find no clear explanation of why it is evoked here. Though the term is in the title, it is mentioned in the text (discussion) only 3 times – 2x when numbers of bars are compared between the groups compared whose affinity is unclear. At the very least it should be dropped from the title. I do not find clear evidence for (between lineage) heterochrony in the statements about numbers of gill bars.

The concept of 'deep homology' is also brought into the mix but is not explained and there is no explicit explanation (connecting the dots) of how it applies to stylophorans. If deep homology is involved what specifically are the authors thoughts on how it applies to their study organisms?

Another comment is that tentaculate and lamellar structures comprised of few to many parallel cylinders each lined with rows of cilia are common throughout many metazoan phyla and are assumed to have arisen independently many times. Motile cilia that move water for respiration or feeding work best when placed on epithelial cylinders extended out into cavities or on cylinders extended from external body surfaces. It is possible the calcified gill bars of stylophorans are an example of convergence without deep homology. The present multidimensional study targets hemichordates and cephalochordates that are favored to compare with the fossil stylophorans, and tosses in a few fossil echinoderms whose bar-like structures are unlikely structural homologs. How these data stack up against other 'water moving' (suspension feeding) organs is not considered.

In my opinion this study lacks the breadth and depth of data to support the discussion. However, the images and quantitative data are excellent contribution to a more specialized journal.

Decision letter (RSPB-2022-0258.R0)

11-Apr-2022

Dear Dr Álvarez Armada

I am pleased to inform you that your manuscript RSPB-2022-0258 entitled "Heterochrony and parallel evolution of echinoderm, hemichordate and cephalochordate internal bars" has been accepted for publication in Proceedings B.

The referee(s) have recommended publication, but also suggest some minor revisions to your manuscript. Therefore, I invite you to respond to the referee(s)' comments and revise your manuscript. Because the schedule for publication is very tight, it is a condition of publication that you submit the revised version of your manuscript within 7 days. If you do not think you will be able to meet this date please let us know.

When submitting your revision please upload a file under "Response to Referees" - in the "File Upload" section. This should document, point by point, how you have responded to the reviewers' and Editors' comments, and the adjustments you have made to the manuscript. We also require a copy of the revised manuscript showing track changes to be uploaded.

4) Data accessibility section and data citation

It is a condition of publication that data supporting your paper are made available either in the electronic supplementary material. Authors must complete the 'data accessibility' section in the submission system. This should list the database and accession number for all data from the article that has been made publicly available, for instance:

NB. From April 1 2013, peer reviewed articles based on research funded wholly or partly by RCUK must include, if applicable, a statement on how the underlying research materials – such as data, samples or models – can be accessed.

[http://datadryad.org/submit?journalID=RSPB&manu=\(Document not available\)](http://datadryad.org/submit?journalID=RSPB&manu=(Document%20not%20available)) which will take you to your unique entry in the Dryad repository. If you have already submitted your data to dryad you can make any necessary revisions to your dataset by following the above link.

Please include the Dryad DOI in the Data Accessibility section and reference in the paper's bibliography.

Please see our Data Sharing Policies (<https://royalsociety.org/journals/authors/author-guidelines/>).

6) A media summary: a short non-technical summary (up to 100 words) of the key findings/importance of your manuscript.

Sincerely,
Dr Locke Rowe
mailto: proceedingsb@royalsociety.org

Associate Editor

Comments to Author:

Your MS has been seen by two experts in the field, however, while one reviewer supports the application the second doesn't. I do think that some of the comments from reviewer #2 are valid e.g. the uncertainty position of stylophorans should be discussed as well as the possible convergence of the calcified gills bar. These criticisms should be addressed in the MS before I can recommend it for publication.

Reviewer(s)' Comments to Author:

Referee: 1

Comments to the Author(s).

Dear authors,

I have read again the MS and response to referees, and you have addressed all my concerns in a satisfactory way. I believe this would be an important paper for Early Echinoderms Evolution with wider implications for the understanding of deuterostome origin. New morphological data is important and statistical approach is novel for the field. Congratulations!

Best wishes,
Samuel

Referee: 5

Comments to the Author(s).

This quantitative study of fossils of extinct stylophorans (*Lagynocystis* and *Jaekelocarpus*), widely accepted as deuterostomes, utilized sophisticated technology to image and reconstruct internal anatomical structures thought to be involved in suspension feeding or respiration. While these have been called gill bars by earlier workers. The actual placement of the group as echinoderms or early chordates is a matter still unsettled in the literature but is key to the overall conclusions. The disputed affinity is not mentioned. Nevertheless, the current study considers them to be early echinoderms and presents quantitative measurements of dimensions of these gill bars and compares them with gill bars of extant hemichordates and cephalochordates.

Additional datasets generated from “bar-like elements” thought to be respiratory structures in 2 other definitive, but extinct, echinoderm groups (blastoids and rhombiferan) are added for additional comparison.

The images of reconstructions of the stylophoran *Lagynocystis* are exquisite (but too small) and add to earlier, similar images (Dominguez et al) of the other stylophoran (*Jaekelocarpus*) used in the dataset.

The multidimensional analyses show that dimensions of gill bars in stylophorans clearly and strongly overlap those of the chordate groups but not those of the echinoderm groups. These data support morphological homology of the gill bars of stylophorans and chordates. They could also be convergent structures.

Based on this dataset and the assumption that stylophorans are echinoderms the manuscript argues that “the latest common ancestor of echinoderms, hemichordates and chordates was a bilaterally symmetrical worm with pharyngeal openings, with these characters lost in echinoderms” (MS Abstract, lines 23-25). The argument is complex and hard to follow as presented. Several current developmental phenomena/processes (heterochrony and deep homology) are invoked to rationalize the inference that gill bars should be present in the last common ancestor of chordates and echinoderms.

I am a reviewer of what appears to be a thoroughly reviewed and revised MS; comments from 4 other reviewers and responses of authors were included in my “review package”. While I’ve read these comments and responses I do not have the time or expertise to arbitrate the revisions. I will give a few comments on my reaction to the current MS.

The images and datasets are important and valuable contributions. The images in the supplementary material are also important for context. Perhaps they should be combined with those of the main text (and those in Dominguez et al) and made into a larger manuscript. (I think a more expansive manuscript directed to a more specialized audience would have a greater, longer lasting impact (than the present MS) on thinking about gill bars and how stylophorans may have worked).

The phylogenetic position of stylophorans is a key assumption and not sufficiently discussed in the MS. So the whole argument is “If stylophorans are echinoderms then” What if they are early chordates? Related is that stylophorans studied here are not the earliest of their clade and the gill bars described here are not reported from earlier group members. That’s curious and not really addressed.

“Heterochrony” is defined as a change in timing of expression of a trait relative to that in a related lineage. I find no clear explanation of why it is evoked here. Though the term is in the title, it is mentioned in the text (discussion) only 3 times – 2x when numbers of bars are compared between the groups compared whose affinity is unclear. At the very least it should be dropped from the title. I do not find clear evidence for (between lineage) heterochrony in the statements about numbers of gill bars.

The concept of ‘deep homology’ is also brought into the mix but is not explained and there is no explicit explanation (connecting the dots) of how it applies to stylophorans. If deep homology is involved what specifically are the authors’ thoughts on how it applies to their study organisms?

Another comment is that tentaculate and lamellar structures comprised of few to many parallel cylinders each lined with rows of cilia are common throughout many metazoan phyla and are assumed to have arisen independently many times. Motile cilia that move water for respiration or feeding work best when placed on epithelial cylinders extended out into cavities or on cylinders extended from external body surfaces. It is possible the calcified gill bars of stylophorans are an example of convergence without deep homology. The present

multidimensional study targets hemichordates and cephalochordates that are favored to compare with the fossil stylophorans, and tosses in a few fossil echinoderms whose bar-like structures are unlikely structural homologs. How these data stack up against other 'water moving' (suspension feeding) organs is not considered.

In my opinion this study lacks the breadth and depth of data to support the discussion.

However, the images and quantitative data are excellent contribution to a more specialized journal.

Author's Response to Decision Letter for (RSPB-2022-0258.R0)

See Appendix B.

Decision letter (RSPB-2022-0258.R1)

19-Apr-2022

Dear Dr Álvarez Armada

I am pleased to inform you that your manuscript entitled "Heterochrony and parallel evolution of echinoderm, hemichordate and cephalochordate internal bars" has been accepted for publication in Proceedings B.

Your article has been estimated as being 8 pages long. Our Production Office will be able to confirm the exact length at proof stage.

Data Accessibility section

Open Access

Paper charges

Sincerely,
Proceedings B
mailto: proceedingsb@royalsociety.org

Appendix A

Associate Editor

Board Member: 1

Comments to Author:

Four experts in the field have reviewed your manuscript. While overall, they find the topic of the work interestingly, they have identified several weaknesses. These critiques concern key aspects of the paper, such as the evidence supporting the heterochrony and parallel evolution. Considering the reviewers' comments, I cannot recommend the manuscript for publication.

Many thanks for taking the time to consider our manuscript. We appreciate the input of the editor and four referees, and we have taken this into account when rewriting the manuscript. We believe this revised version is much stronger as a result of these comments. We note that all four reviewers regard the manuscript as interesting and important, and their main criticisms were about how our findings were presented and justified. We have made changes to the main text and figures to better support our interpretations, in particular the evidence for heterochrony and parallel evolution (details below). We have also added additional information, such as schematic drawings and detailed descriptions of the anatomical characteristics of the targeted structures in all taxa to facilitate the comprehension of a wide diversity of readers, including non-experts in the topic of this study. We therefore hope this revised version will prove acceptable for publication, albeit after further evaluation.

Reviewer(s)' Comments to Author:

Referee: 1

Comments to the Author(s)

This paper analyzes the presence of gill slits in an enigmatic group of early echinoderms known as stylophorans. The authors compare these structures using state of the art techniques with other deuterostomes and also with respiratory-like structures from other echinoderms.

The paper is well written, well supported and it is a novel approach with important phylogenetic implications for the understanding of early deuterostome evolution. I think this paper is an important contribution for Proceedings B. Nevertheless, I have several points that should be considered by the authors to strengthen their hypothesis.

We thank the referee for the very positive comments on our study and its suitability for publication in Proceedings B. We are also very grateful for their suggestions on how to strengthen our main arguments, which we have incorporated into the revised manuscript (details below).

The presence of gill slits in early echinoderms (especially in stylophorans) has been long debated from almost a century. There are two competing hypothesis, one suggesting that openings in stylophorans are respiratory structures and other suggesting that they were pharyngeal openings. In this aspect the approach of comparing these structures with those seen in hemichordates and cephalochordates using CT-scan and morphometrics is really interesting. My main concern on the pore-like structures seen in stylophorans is that earliest members of the group lack or not preserve the bars seen in *Lagynocystis* for example. The authors tackle this problem in lines 252-258; but I think they should expand such discussion. Could be these structures non-mineralize in early echinoderms, being made of cartilage? Why the external morphology of these structures vary so much in different stylophoran groups? etc.

We agree with the referee that these are interesting points that merit further consideration, and we have expanded our discussion accordingly. As noted in our original submission, while collagen is not known to be a precursor of a biomineralized skeleton in any extant invertebrates, it is a precursor of apatite in vertebrate biomineralization. This raises the possibility that the earliest echinoderms might have possessed collagenous gill bars that are not preserved in any known fossils due to the lower preservation potential of this tissue relative to echinoderm-type calcite, with calcification of gill bars only occurring (independently) in *J. oklahomensis* and *L. pyramidalis*. While gill bars inferred to be originally composed of collagen have been reported in some Cambrian deuterostomes (e.g. the hemichordates *Spartobranchus* and *Oesia*; see Caron et al. 2013; Nanglu et al. 2016), these fossils are almost exclusively known from Burgess Shale-type Lagerstätten, from which stylophorans and other putative stem echinoderms are absent. As to why the external morphology of pharyngeal openings apparently varies so much among stylophorans, it could be due to the chaotic way in which stylophorans are grouped. This is not uncommon among other groups of Palaeozoic echinoderms, in which most often the external morphology of the structures is quite variable but is still used to delineate species with usually mixed results. However, this is a question that we hope to address in a future study analysing these structures across the group more broadly. The discussion has been expanded to address these important points [lines 363–392].

Lines 65-66 – The authors indicate that stylophorans comprise “two clades” but it is largely demonstrated that they don’t. As far as I know from recent phylogenies cornutes are paraphyletic respect to mitrates. They are two clear traditional groups but phylogenetically speaking they are not two clades.

We agree, and have revised the main text so that cornutes and mitrates are no longer referred to as clades, but instead as “traditional groupings” [line 80].

Comparison with stylophoran openings and those seen in other echinoderms (like blastoids and rhombiferans) is really welcome. First, I would like to emphasize that different types of respiratory structures in echinoderms are likely being analogous rather than homologous. So, in the supposed case that stylophoran openings were respiratory structures, they must be independently acquired in such group.

We agree. The main text has been amended to reflect this [lines 90–92].

Otherwise, I find very interesting the morphometric analysis showing that openings in stylophorans, hemichordates and cephalochordates cluster together in the figure 2. Paul (1968, PALAEOLOGY vol. 11) reported a large study of respiratory structures in cystoids with measures from different taxa (table 1). I just wonder if the authors are able to include such data in the analysis to reinforce the differences between respiratory structures and gill slits.

We carefully considered whether it would be possible to incorporate any of the data from Paul (1968) into our analysis. The measurements reported in Paul (1968) coincide with three out of the four measurements used in our dataset (i.e. length, width and spacing); however, the values given by Paul represent averages for each of the dimensions reported. Our dataset consists of individual raw values of each dimension measured, standardised against the total length or diameter of the pharynx/thecal cavity. As a result, Paul’s (1968) dataset is unfortunately not compatible with our analysis. We have added a new section in the Supplementary Text file titled “Comparisons with previous data” to reflect where our data stands in comparison with previous studies.

Congratulations on such interesting work,
Samuel Zamora

Referee: 2

Comments to the Author(s)

The evolution of pharyngeal arches is of fundamental importance to the early evolution of deuterostome animals - indeed, as the authors point out, some form of pharyngeal pore (and, by extension, arches the sit in between these pores) is the only unambiguous synapomorphy of deuterostome animals. It is now well accepted that the pharyngeal pores of cephalochordates and hemichordates are homologous (at the level of endodermal pouch derivatives) with the gill slits of vertebrates. Current deuterostome phylogenies would therefore suggest that the absence of such pores in extant echinoderms is a product of evolutionary loss. Consistent with this scenario is the presence of putative pharyngeal bars in the stylophorans, a group of stem-echinoderms. In this manuscript, the authors use comparative analysis of gill bar size and spacing in two stylophorans and two extant deuterostomes (a cephalochordate and a hemichordate) to test the homology of these structures. They find that the gill bars of stylophorans are more similar (in morphospace) to those of cephalochordates and hemichordates than to the internal bars of another group of fossil echinoderms (the blastozoans). They conclude that stylophoran gill bars are homologous to those of cephalochordates and hemichordates, and make various speculations about heterochrony and parallel evolution in gill bar evolution.

I think that there is a valuable contribution in this manuscript - this sort of quantitative analysis of stylophoran gill bars has, to my knowledge, not been performed previously. However, I think that this article will be much better suited to a more specialist journal, and will require some additional fleshing out of the content before further consideration.

We appreciate the referee's comments, but strongly disagree our work would be better suited to a more specialist journal. As highlighted by the reviewer, we present the first rigorous quantitative analysis of gill bar-like structures across a range of extinct and extant deuterostomes, which represents a major step forward in the field (as noted by all four referees). The results strongly suggest that internal structures preserved in two enigmatic fossil echinoderms (*J. oklahomensis* and *L. pyramidalis*) are homologous to the gill bars of extant deuterostomes. This has significant implications for understanding the evolution of pharyngeal openings in echinoderms and allows us to propose a new hypothesis for the evolution and development of internal bar-like structures across the phylum, challenging existing ideas about the origin and evolution of major morphological characters. Consequently, our findings will be of broad relevance to biologists interested in deuterostome evolution, comparative biology, evolutionary genetics and palaeontology. We are grateful for the opportunity to correct errors and clarify ambiguous points highlighted by the referee (see below), and firmly believe our revised manuscript is therefore well suited for publication in Proceedings B.

1. I think that a more nuanced Introduction to the pharyngeal structures of deuterostomes is needed in a paper like this. Deuterostome pharyngeal bars and gill pores are an excellent example of the hierarchical nature of homology: the pores form from endodermal outpockets (which are homologous to those of vertebrates, and which share gene expression features with those of vertebrates - particularly in relation to the conserved syntenic block of "pharyngeal" transcription factors mentioned in the text). And by extension, the tissue in between neighbouring pores (the pharyngeal arch) is also homologous - though there are additional tissues inside the pharyngeal arches of vertebrates that are not found within the pharyngeal arches of cephalochordates and hemichordates. This is touched upon in the Discussion (though not with sufficient detail), and also appears in the final figure (though I would argue, again, with insufficient treatment in the text). I

think that the introduction should probably contain at least some brief discussion of how these pharyngeal structures form in hemichordates, cephalochordates and vertebrates, and the level(s) at which these structures are or are not homologous.

This is a good point. To address this, we have added more detail on the anatomy and development of pharyngeal structures in the deuterostomes included in this study to the introduction [lines 93–111] and the main text [lines 266–279]. In addition, we have added new schematic diagrams to Fig. 1 (c, e, g, j) to better illustrate the morphology of deuterostome pharyngeal openings. However, we believe the discussion of pharyngeal bars in vertebrates is beyond the scope of our study.

2. Also, the blastozoans just appear at the end of the Introduction, and it isn't entirely clear why. What are these animals, and why do they make a relevant comparison with stylophorans or other deuterostomes? This might be obvious to an echinoderm palaeontologist, but I don't think that others would necessarily immediately know the relevance of this group.

Blastozoans are an extinct group of echinoderms characterized by specialized respiratory structures, which can take the form of elongate bar-like folds and pores that are broadly similar to the internal bars seen in *J. oklahomensis* and *L. pyramidalis*. Consequently, any rigorous analysis of the internal bars of stylophorans must consider their similarity to bar-like structures in blastozoans, as well as the gill bars of extant deuterostomes. We have added a sentence to the introduction to better explain why blastozoans are a relevant comparison to stylophorans [lines 90–92].

3. I am not a palaeontologist, and I only know about stylophorans from reading a few papers. I found it extremely difficult to understand the anatomy of these pharyngeal structures (and also of the internal bars of blastozoans) from Figure 1 and the supplemental figure. I think that it would be extremely helpful for non-palaeontologist/non-stylophoran experts if the authors could provide anatomical schematics or drawing to accompany their CT scans. This would provide some important organismal context for the animals that are being discussed here.

We thank the reviewer for this excellent suggestion, which we agree would be very helpful for readers unfamiliar with the anatomy of these taxa. To address this, we have added new schematic diagrams to Fig. 1 (c, e, g, j) to better illustrate the morphology of the internal bars and bar-like structures that are the focus of our study.

4. Also, is it not important to consider deformation/taphonomy when comparing quantitative morphology of fossil and extant structures? Especially when these structures differ in the nature of their tissues (soft vs. mineralised), and when only a very small number of individuals are being assessed?

The referee is correct that taphonomy must always be considered when dealing with fossil material. All the fossils included in our analysis are three-dimensionally preserved. Specimens of *L. pyramidalis* are preserved as largely articulated, essentially uncompressed moulds in siliceous nodules, while *J. oklahomensis* and the fossil blastozoans are preserved three-dimensionally as recrystallized calcite. In all our fossil material, the internal bars and bar-like structures were originally calcified (as opposed to being soft structures that could be more susceptible to post-mortem deformation), and there is no external evidence of deformation of the skeleton. Based on these observations and through detailed comparison with hundreds of other museum specimens, we are very confident that the specimens accurately represent the original hard part morphology of the living animals. We have added some text explaining this to the electronic supplementary information.

5. While the statistical analysis performed seems sound, I have some concerns that the main points

of the paper (i.e. heterochrony and parallel evolution of pharyngeal bars) are more speculation and hypothesis - i.e. the authors speculate about these patterns of evolution, rather than providing an abundance of new evidence for them. I don't think that there is anything wrong with this necessarily, but I do think that this is one of the aspects of the manuscript that makes it more suitable for a specialist journal, and perhaps less suitable for a general science journal like Proc. B.

We disagree with the suggestion that our main findings are too speculative. The new hypothesis we propose for the evolution of gill bars and bar-like structures across echinoderms is supported by the results of our rigorous quantitative analysis, which clearly showed very close similarity between the internal bars of the stylophorans *L. pyramidalis* and *J. oklahomensis* and the gill bars of extant chordates and hemichordates. This is in stark contrast to the blastozoans, which are characterised by bar-like structures with very different dimensions to the other extinct and extant taxa.

With respect to heterochrony, we use this term with its modern meaning - a relative change in timing of one morphological trait with respect to another. See our responses to the last comment from referee #4 and comment #7 of this referee. We have also expanded the discussion around this topic [lines 254–279]

There is a passing mention, for example, that the absence of secondary gill bars in stylophorans is a product of heterochrony - but isn't it possible that the resolution of the anatomy from these CT scans isn't enough to distinguish between primary and secondary gill bars (if the latter are actually present)?

This is very unlikely to be the case. In adult extant deuterostomes, secondary gill bars range in size from 500–2000 μm in length, 40–70 μm in width, 20–120 μm in depth and 40–90 μm in spacing. The resolution of our microCT scans of the stylophorans was between 3–20 $\mu\text{m}/\text{pixel}$, which would be sufficient to reveal any secondary bars if they were present.

Or rather than heterochrony (which implies a shift in the relative timing of developmental events through evolution), couldn't stylophorans have just lost secondary gill bars? With heterochrony in the title, I was expect a more fulsome discussion around this point, but I felt that this wasn't thoroughly treated in the text.

We agree with the referee that it is possible the secondary gill bars could have been secondarily lost in the stylophorans *L. pyramidalis* and *J. oklahomensis*. However, this would not in itself explain why the number of internal bars is so much lower than that seen in the adults of all extant deuterostomes. We therefore consider that heterochrony represents the most plausible scenario for the evolution of the gill bars in *L. pyramidalis* and *J. oklahomensis* [lines 254–279], even if it is impossible to prove this beyond all doubt. We have inserted some additional caveats into the text to emphasize that some degree of uncertainty surrounds this interpretation [lines 390–398].

6. Again, more nuance need in the Discussion re: homology. How is it known that the stylophoran gill bars were composed of calcite?

The internal bars in *L. pyramidalis* and *J. oklahomensis* are preserved in an identical manner to all other parts of the fossil, which are unambiguously interpreted as having been originally composed of calcite owing to the preserved plate morphologies and the presence of stereomic microstructure in other stylophorans (e.g. Clausen & Smith 2005) [lines 281–308]. In *L. pyramidalis*, the original skeleton had dissolved to leave behind a mould, whereas in *J. oklahomensis*, the skeleton was replaced by recrystallised calcite during diagenesis. In all cases, no soft parts are preserved in the fossils, and so if the internal bars had a different composition to the skeleton (e.g. collagen) we

would not expect them to be preserved as fossils. We discuss the preservational characteristics of the fossil samples in the main text [lines 281–308] and the electronic supplementary information.

And discussion of hemichordate pharyngeal bars of hemichordates being "composed of extracellular cartilage" is imprecise - the bars do contain a collagenous extracellular matrix, which appears to be secreted by the pharyngeal endoderm. But this does not equate with cartilage - again, homology at the level of general expression of SoxE/Col gene regulation does not mean homology at the level of tissue. These issues need to be discussed more thoroughly.

We thank the referee for highlighting this. We have corrected the text with "cartilage" replaced by "collagen" [line 286].

7. "We suggest that differences between the internal gill bars in stylophorans and living deuterostomes, such as the number and morphology of the bars, are the product of heterochronic evolution." Why?

The secondary gill bars of cephalochordates and enteropneusts are not present in *L. pyramidalis* or *J. oklahomensis*, therefore the relative development of secondary bars, with respect to primary bars, is slowed: or *heterochrony* (it's also *heterometry* – the quantity of primary to secondary bars – but this term is less well known). The total number of bars relative to adulthood is also *heterochrony*. There are fewer internal bars in the stylophorans (~25 in *L. pyramidalis* and 8 in *J. oklahomensis*) than in the cephalochordate (~250) and enteropneusts (~130 in *Schizocardium* sp. and ~154 in *Balanoglossus* sp.). The development of cephalochordate gills also represents *heterotopy* – because the relative *position* of the first developing pores are asymmetric – but this is well known and not the topic of our study. See expanded discussion on this topic [lines 254–279].

8. "In contrast, we infer that bar-like elements of the respiratory systems in blastozoans, which are superficially similar to the internal bars of stylophorans and some extant deuterostomes, but statistically dissimilar in terms of their size and shape, evolved independently, perhaps by parallel evolution." Parallel evolution (i.e. independent evolution of morphologically similar structure, due to common underlying developmental mechanisms) of gill bars in blastozoans is untestable. So fine to speculate, but I don't think there is much to say about this other than "could be...".

Although we believe this is the most likely scenario based on the conservation of pharyngeal gene clusters across deuterostomes, including echinoderms, we accept that it is impossible to unequivocally demonstrate parallel evolution underlies the superficially similar internal bar-like structures observed in fossil blastozoans. As a result, we have added some caveats to the text as suggested by the referee [lines 390–398].

Overall, I think that a more fulsome and nuanced analysis of pharyngeal bar morphology (integrating anatomy and morphometry), with discussions of homology, and possible scenarios of evolution could be a really nice contribution to the literature on this subject - but I think that it requires expansion of several of these points highlighted above, and is more suitable for a specialist journal.

We strongly feel the research presented in our manuscript is well suited for Proceedings B, as highlighted by referee 1. As discussed above (and noted by the other referees), our findings will be of interest to a wide range of biologists working in different fields, such as evolutionary biology, palaeontology and developmental biology, among others. We have expanded our discussion of homology and evolution, including some more caveats as requested by the referee. We therefore believe our revised manuscript is appropriate for publication in Proceedings B.

Referee: 3

Comments to the Author(s)

The origins and evolutionary history of deuterostomes is a fascinating topic with much left to be understood. Developing a firm understanding of the morphological features of early groups is integral to illuminating the story of early deuterostome evolution. This paper explores the presence of gill bars in several early deuterostome taxa, providing inferences about what the presence of certain characteristics can tell us about the evolutionary history of these anatomical features and deuterostomes overall. The report also provides openly available high-quality micro-CT imaging data of several deuterostome taxa. There are a few issues to be addressed prior to publication.

General comments:

1. A number of similar anatomical terms are used to describe the characteristics of interest. For instance, in the abstract alone, we have: pharyngeal openings, pharyngeal gill slit-like structures, gill bars, internal bars, internal bar-like structures, bar-like elements. It would be helpful to get some more information regarding the definitions and differences between these terms. Are these all technical terms referring to different features across organisms? Are these technical terms referring to similar structures in different organisms? Do these represent a mixture of technical anatomical names and descriptive terms? Are some of these terms synonymous? If not, what is different about how these structures function in living organisms? I would suggest providing some more context about how these structures have been described before and what prior studies have suggested in terms of homologies. A diagram would be fantastic.

We have simplified the wording used in the abstract and through the main text as suggested. The terms “pharyngeal gill bars” and “gill bars” refer to the collagenous, elongated internal structures found in the pharynges of some modern deuterostomes; “pharyngeal openings” refers to the outlets in the pharynges, which are surrounded by the collagenous gill bars in some modern deuterostomes; “internal bars” refers to calcified, elongated bars found in the internal body chamber (or theca) of the stylophorans *J. oklahomensis* and *L. pyramidalis*; “internal bar-like structures” refers to the calcified folds in fossil blastoids and pores in fossil rhombiferans both presumably functioning in respiration. We have added labels to the amended versions of Fig. 1, Supplementary Fig. 2 and expanded the supplementary text with a “Terminology” section to clarify this.

2. As it is written, it is unclear how the data provided here can be used to suggest that the internal bar structures in stylophorans are homologous with those in extant deuterostome taxa, as it seems as though these features are present in only two stylophorans that are both relatively derived.

We have expanded our discussion of this in the main text [lines 362–407]. See also our response #1 to Referee 1.

It is also not clear how the specific measurements of these anatomical features performed address questions regarding how these structures evolved and are related to one another. Is there a precedent for this, or a gap in the literature that measuring these specific components is solving?

Several previous studies (e.g. Jefferies 1973; Dominguez et al. 2002) have compared the internal bars in the stylophorans *L. pyramidalis* and *J. oklahomensis* to the gill bars of extant deuterostomes. This work relied heavily on qualitative comparisons to infer homology (i.e. similarity due to shared ancestry). We present the first quantitative comparisons of these internal structures, which is a more rigorous way of assessing their similarity. Our statistical analyses demonstrate that the

similarities between the bars of the two stylophorans and the extant deuterostomes are strongly statistically significant. Coupled with previous remarks on the general morphological similarity of stylophoran internal bars to deuterostome gill bars (Jefferies 1973; Dominguez et al. 2002), this strongly implies homology of these structures. This is also consistent with recent molecular genetic evidence indicating homology of deuterostome pharyngeal openings.

3. Are there functional implications about the physiology of stylophorans or blastozoans that can be inferred from measuring the dimensions of their internal bars?

We prefer not to speculate too much about the function of internal structures in extinct echinoderms as our analyses do not directly inform on this. However, the close similarity between the dimensions of the internal bars in stylophorans and deuterostome gill bars, in particular their spacing, could point towards the presence of cilia covering the bars; in modern deuterostomes, the gill bars are lined with cilia, which partly fill the space between the bars and are important for pumping water. The same may have been true for stylophorans, albeit there is no direct evidence of this preserved in the fossils. This is now mentioned in the main text [lines 342–347]. Future work using computational fluid dynamics to simulate water flow past these structures could help better constrain the functional morphology of these enigmatic extinct organisms.

Specific comments:

Line 46: What is the function of pharyngeal openings in the living organisms that have them?

Pharyngeal openings are outlets in the pharynx, which play important roles in feeding or respiration in extant deuterostomes. This is now noted in the introduction [lines 56–57].

54: What is the difference between gill pouches and pharyngeal openings? Gill bars and pharyngeal gill slits? A diagram, more context, or definitions would help here and elsewhere (see first general comment).

See response to comment 1 from the referee above. Gill pouches are a series of protuberances in the pharynx seen in embryonic chordates and hemichordates, from which the pharyngeal gill bars (and gill pores created by the displacement of tissue through the growth ventrally of the bars) arise later in development. Gill slits are the openings to the gills (a single opening in between two gill bars) in vertebrates, homologous to the pharyngeal openings in all deuterostomes. To make this clearer, we have simplified our wording throughout the manuscript. We have also modified Fig. 1, Supplementary Fig. 2 and expanded the Supplementary Text adding a new section titled “Terminology” to better describe pharyngeal structures across deuterostomes.

Line 64: What makes stylophorans particularly significant? Does it have to do with their anatomy or their position within the deuterostome phylogeny?

Stylophorans are characterized by an aberrant asymmetrical body plan; their phylogenetic position is debated, but they have been suggested to represent early diverging stem echinoderms. In addition, they are the only fossil echinoderms that preserve direct evidence of putative pharyngeal openings. They are therefore crucial for understanding the evolution of pharyngeal structures in echinoderms. This is explained in the introduction [lines 84–88].

Line 70: Are there other stylophorans that have internal structures interpreted as gill bars, or is it just these two?

No other stylophorans are known to have possessed internal bars; however, it is uncertain if this reflects a difference in their original composition (e.g. collagenous bars would be less likely to preserve in fossils than calcite ones) or a genuine absence. See response #1 to Referee 1 and our expanded discussion [lines 362–392].

Line 72: What are gill bars specifically? How are these different/similar to pharyngeal openings? Can you talk about the evolutionary significance of an echinoderm with these kinds of features?

Gill bars act as skeletal support for ciliated tissue in the pharynx, whilst pharyngeal openings are outlets for the pharynx (see response to the first specific comment in this review, also [lines 56–60]). Stylophorans are evolutionarily significant due to their unique combination of morphological characters, which mix features from modern echinoderms with hemichordates and cephalochordates; the presence of gill bars in stylophorans would therefore have important implications for reconstructing the sequence of acquisition of key deuterostome characters. This is noted in the main text [lines 415–420].

Line 77: What specific evidence would provide support that these are gill bars?

To test this hypothesis, we measured and described the size and shape of the internal bars in the two stylophorans and quantitatively compared this to (1) gill bars of extant hemichordates and cephalochordates and (2) morphologically similar bar-like elements of the respiratory systems in three blastozoans. We argue that statistically significant similarity in terms of the dimensions of these bars would support their homology.

Line 80: Why were these taxa selected for comparison?

We selected cephalochordates and enteropneust hemichordates to include in our study because both taxa possess pharyngeal gill bars and they have previously been compared to stylophorans (Jefferies 1973 and Dominguez *et al.* 2002). This is noted in the main text [lines 84–88].

Line 82: Why were blastozoans selected for comparison?

See response to comment #2 by Referee 2. Also see [lines 90–92] in main text.

Fig. 1: This figure should be larger and higher resolution. I would also recommend putting Supp Fig 1 in with the images of the larger organisms in Fig 1/Supp Fig 2 together to provide context for where the close-ups from S1 are in the organism.

Fig. 1 has been enlarged and modified with new schematic diagrams to better illustrate the morphology and location of the internal structures represented in Supplementary Fig. 1. Additionally, explanatory labels have been added to Supplementary Fig. 2.

156: What is the basis of the assumption that homology/functional similarities can be inferred from the measured characteristics?

Quantitative comparisons of morphology of the anatomical characteristics of fossil organisms are the most accurate way to assess similarities among structures and infer homology [lines 84–90; 222–237]. Also, see response to the comment below.

Is there a justification functionally/phylogenetically as to why these dimensions of the internal bar/bar like structures was selected for measurement specifically?

We believe quantitative measurements of bar length, width, depth and spacing are the most appropriate dimensions to fully summarise the morphology of the pharyngeal bars in modern deuterostomes, internal bars in fossil stylophorans and bar-like structures in fossil blastozoans. Combinations of these measurements have been used before to describe the morphology of these structures in hemichordates (Cameron 2002; Vo et al. 2019), cephalochordates (Rähr 1982), stylophorans (Dominguez et al. 2002), blastoids (Bauer et al. 2017) and cystoids (Paul 1968). Additionally, the spacing between bars could be an indication of the presence of cilia in live (Dominguez et al. 2002) [lines 342–347].

Is there a significance in terms of indicating homology or function?

See answer to comment on line 156 below.

What is the morphological disparity of these measured characteristics between other living organisms with gill bars?

There are numerous similarities among deuterostome gill bars (see e.g. Ruppert 2005), however a likely interesting outcome, a task of this calibre lies beyond the scope of the current research.

What would finding morphological differences/similarities between taxa actually mean?

In palaeontology, morphology is commonly the only clue we have to analyse the phylogenetic relationships across groups. Differences and similarities of morphological characters are the key tool for elucidating patterns of character evolution in extinct and extant taxa. Additionally, in this case we also have a strong genetic support indicating the cluster of genes around gill pouches is present in several embryonic deuterostomes (including echinoderms), which suggests that this cluster must have been expressed at some point in echinoderm evolution.

Line 183: Is there an established threshold for when similarities between structures would suggest that they're homologous or not?

To the best of our knowledge, the threshold for determining the homology of morphological characters has never been quantified. Our analyses provide strong statistical support for homology. Our data has shown that there is a strong statistically significant similarity between morphological dimensions, such as length, width, depth and spacing, of the internal bars of extinct stylophorans and the gill bars in hemichordates and cephalochordates. There is also a strong relationship between the dimensions of these structures and the total body length of the animal, a relationship that is not maintained in the blastozoans analysed. Furthermore, there are statistically significant morphological differences between the internal bar-like elements in the blastozoans and the stylophorans plus extant hemichordates and cephalochordate [lines 222–237]. Also see answer to comment #2 from referee 3.

Line 183: Are these the only two stylophorans with gill bars? If so, you mention that they're relatively derived groups (line 252)- how could these structures then be homologous with other non-stylophoran taxa? It's difficult to accept that that's the most parsimonious conclusion to draw, especially due to the differences between the gill bars in stylophorans and living deuterostomes as you mention (line 259). If this claim is going to be asserted, an ancestral state reconstruction should be performed demonstrating that this is the most likely case.

See our response to Referee 1 above. The primary aim of our study was to evaluate the evidence for homology of the internal calcite bars found in two fossil stylophorans and the gill bars in hemichordates and cephalochordates, rather than to reconstruct the pharyngeal structures of early echinoderms as a whole. Future work will include analyses such as phylogenetic inference and ancestral state reconstruction. Nevertheless, we have expanded our discussion on stylophoran internal bars, including some consideration of possible preservation biases that would explain their apparent absence in other fossil taxa [lines 363–392].

Line 194: What is the functional/evolutionary significance of secondary gill bars?

The downwards growth of the secondary bars divides one gill slit into two slits, thereby approximately doubling the surface area for the capture of particles.

Line 196: An order of magnitude seems like a big difference in the number of internal bars between these groups- is there functional significance there?

We cannot point to any evidence that would make us think there is any functional significance to this difference. Neither of the stylophorans seem to have developed secondary bars, whilst both hemichordates and cephalochordates have, and this essentially doubles the total number of bars in the pharynx. Also, between *Schizocardium* sp. and the cephalochordate are over 100 gill bars of difference and the functionality does not vary.

211: If these are homologous structures, what implications does this have for the relationship between bone/stereom development?

As noted in the main text [lines 308–311], invertebrate collagen is not known to precede mineralisation of any type; however, in vertebrates collagen is well documented to be the predecessor of mineralisation and the formation of bone. We infer that collagenous gill bars were present in other early echinoderms, and these were calcified independently in *J. oklahomensis* and *L. pyramidalis*. This is consistent with the suggestion that a common developmental genetic mechanism underpinned biomineralization in chordates and echinoderms. See our answer to comment #1 from Referee 1.

Line 180/Fig 2: The approach is interesting and the data collected and presented is definitely worthwhile, but it is difficult to be convinced that the dimensions measured here are sufficient to determine homologies. Can you provide more justification for this or cite something using the same approach? I've seen this type of analysis used previously to infer functional similarity but not homology necessarily. Are there other things about the physiology of the organism that performing these measurements can tell you about perhaps?

Currently there is extended literature concerning the challenges of using morphology to establish homology (see for example Young 1993 and Vogt *et al.* 2010). Homology has historically been tightly tied to morphology in many studies, particularly those involving extinct organisms, since commonly morphology is the only indication of phylogenetic relationships that we have. See Patterson 1988, where tests of similarity, congruence and conjunction are applied in order to accept or reject homology. Nonetheless, qualitative observations of morphology alone may be insufficient to denote homology, and we argue for the importance of correlating morphological characters in conjunction with another independent datasets such as molecular data to describe homology, since morphology is widely accepted to improve phylogenetic studies. Whilst the focus of our work are the statistically significant similarities between the internal bars found in stylophorans with the gill bars in modern cephalochordates and hemichordates, our interpretation of homology is also supported by previous

genetic analyses, where the transcription factor genes coding for gill pouches in embryonic deuterostomes are conserved across all extant taxa 9albeit not expressed in any living echinoderms).

Referee: 4

Comments to the Author(s)

I have very few specific comments to make, but some general ones.

Specifics:

The figures MUST be enlarged.

Figure 1 has been enlarged and amended as requested (see also response to referee 3 above).

Schizochardium in Table 1 should be Schizocardium.

This has been corrected.

By the way, why are the species identity not known? At least for the extant species this should be remedied.

This corresponds to a new species, which will be described in a separate manuscript (currently in preparation). We therefore refer to the taxon as *Schizocardium* sp. in our study.

In the main text the references to the figures does not match the actual figure number.

We have carefully checked and corrected all the figure citations throughout the main text.

Branchiostoma floridae in Table 1 should be italicized.

Amended.

In line 119: not the "Tomographic datasets" ... were ... reconstructed Rather, the tomographic datasets were used to digitally reconstruct the anatomy of the 10 specimens. Please rephrase.

This has been rephrased as suggested by the referee [line 156].

Semi-specifics:

lines 199-201: during the early development ... begin as a single pair of pots, with subsequent pairs added posteriorly during growth. In Cephalochordates the ontogeny of gill slits is rather complex and starts out with asymmetric pores arranged so oddly, that evolutionary speculations abound. (I attach an old article of mine that has a drawing on page 10. There are much better articles to follow the development of gill slits, yet this is on my laptop now.)

We thank the referee for highlighting the peculiarities of gill slit ontogeny in cephalochordates! We have modified the text to make this distinction clearer [lines 270–273].

line 236: not the bars are involve in the mentioned physiological processes, but the living tissues in the filaments are.

This has been amended [line 344].

lines 252, 253: In my opinion (and I follow Jenner (2006; Unburdening evo-devo: ancestral attractions, model organisms, and basal baloney)), not animals are primitive/derived, but specific characters are. And that only in a certain context, i.e. comparatively.

We no longer refer to any taxa as 'primitive' or 'derived' in the text [line 362].

General:

I found the morphological descriptions in the results section barren. For the fossils there is not enough detail given in terms of their position and relation to other identifiable structures within the fossil to gauge any similarity of the respective fossil structures to gill bars in extant taxa. In fact, at least in my opinion, the results section "Anatomical description" not only delivers very scarcely, what the title promises, but it also does not clearly distinguish between results and interpretation. In fossils, anatomical denominations, such as slits, pores, ambulacra, thecae cavity, hollow folds etc. are already interpretations, because all you have are differences and lines and shapes in a rock. Most often, even hollow spaces or cavities are now filled by rock. Thus, either justify the interpretations in the results section by plausible descriptions and supportive figures, or state clearly, that you follow the published interpretation of another researcher.

Due to space limitations, we included only a short description of the internal bars and bar-like structures in the main text. However, we provide a much more comprehensive description of the key anatomical characteristics of all the taxa included in our study, supported by a supplementary figure, as electronic supplementary material. This is cited in the main text [lines 205–207]. The anatomical description in the electronic supplementary material also includes citations to key references describing and interpreting the morphology of the fossil taxa, which agree with our findings.

The discussion parts about heterochrony, parallel evolution, and deep homology do not convince me. There is neither enough consideration of morphological complexity/specifics to allow for the conclusion of parallel evolution. Nor is there any indication of ontogenetic shifts in the development of any of the structures implicitly proposed to be homologous to allow for the descriptor "heterochrony". A more detailed phylogenetic account would be necessary in that case as well. At least, I would expect to include (some) ascidians, cyclostomes, hagfishes, and gnathostomes in such speculations. I am also puzzled by the brazen interpretation of transcription factor genes as indicative of underlying parallel evolution of bar-like structures. Naturally, nothing is known about the context of these genes in blastozoan development. (As a remark on the side, the fact that these genes are present in echinoderms without any gill bar like structures, shows that these transcription factors can evolve independently from the structures. Consequently, similarity in gene expression would not (necessarily) indicate homology of the resulting structures.)

We found no statistically significant differences in the width or length of the bars among the enteropneusts and stylophorans, or in the spacing of gill bars in the stylophorans, cephalochordates and enteropneusts (Supplementary Table 3) [lines 222–237], whereas these structures show statistically significant differences from the other fossil echinoderms included in our study. We strongly believe that the most interpretation of these results is that the structures are homologous.

With respect to heterochrony, we are not using *heterochrony* in the sense of the traditional definition that is limited to a relative change in the timing of a somatic trait vs. a reproductive one, but rather the modern definition: a relative change in timing of one morphological trait with respect to another. In our case, i) the timing of primary gill bar relative to secondary gill bar development, and ii) the relative number of gill bars in adult animals. Also, see answer to referee 2, comment #7 and expanded discussion [lines 254–279]. A morphological analysis of chordate gills (i.e., ascidians,

cyclostomes, hagfishes, and gnathostomes) is outside the objectives of this study, which aims to better understand and qualitatively compare the gills of hemichordates, echinoderms and cephalochordates. Also, there are no homologues to the gill bars of our study animals with the stigmata of the uniquely tunicate branchial basket.

With respect to *parallel evolution*, we found the form and length of the folds in the blastoids and pores in the rhombiferan to be statistically significantly different from the length of the internal bars in all other groups (Supplementary Table 3). These data are in agreement with specialists of blastoids and rhombiferans, that regard the structures as sufficiently unique to call them 'folds' and 'pores', rather than 'gills'. Where we depart from previous workers is hypothesizing that they are due to *parallel evolution*. That is, the molecular developmental, genetic tool kit was likely conserved across echinoderms (as it is across Ambulacrarians), but the folds and pores evolved independent from ambulacrarian gills. Referee 4 is quite correct here to point out that with respect to gene order in blastozoans, nothing is known (or can be known). The hypothesis that the blastozoan genome showed synteny of pharyngeal genes is based on shared pharyngeal gene synteny in living echinoderms and hemichordates, that was not previously available to specialists of these extinct echinoderm taxa. Ancestral gene order construction is no different than ancestral state construction of gene sequences, or morphological traits, based on living animals. Parallel evolution is the most parsimonious hypothesis based on current evidence, but it is a hypothesis. We acknowledge the possibility that pharyngeal synteny was not shared in blastozoans, but this hypothesis would require additional steps, other developmental genes, or gene orders, that are entirely unknowable. We have added caveats throughout the main text to emphathise purely hypothetical thoughts.

As an unashamed move of self-advertising, I also attach another article, a book chapter, that might be of interest, because it is on the general subject of morphological evolution in deuterostomes. And relatively obscure.

We thank the reviewer for sharing these interesting references, which were very helpful for revising our manuscript.

I do hope this is not too negative but worthy of consideration. If you find it too negative, please kindly ignore it.

Best wishes and sincere regards

Thomas Stach

Appendix B

Associate Editor

Comments to Author:

Your MS has been seen by two experts in the field, however, while one reviewer supports the application the second doesn't. I do think that some of the comments from reviewer #2 are valid e.g. the uncertainty position of stylophorans should be discussed as well as the possible convergence of the calcified gills bar. These criticisms should be addressed in the MS before I can recommend it for publication.

Many thanks for taking the time to consider our manuscript. We appreciate all the comments from the editor and the two referees, which we have taken into consideration when revising the manuscript. We find the comments regarding the uncertain position of stylophorans within deuterostomes and the possibility of convergent evolution of the calcified gill bars in deuterostomes valid, and have answered these comments below and expanded our introduction and discussion sections accordingly. We believe this revised version is stronger and more appealing to both specialist and non-specialist readers.

Reviewer(s)' Comments to Author:

Referee: 1

Comments to the Author(s).

Dear authors,

I have read again the MS and response to referees, and you have addressed all my concerns in a satisfactory way. I believe this would be an important paper for Early Echinoderms Evolution with wider implications for the understanding of deuterostome origin. New morphological data is important and statistical approach is novel for the field.

Congratulations!

Best wishes,

Samuel

We thank the referee for the very positive comments on our study.

Referee: 5

Comments to the Author(s).

This quantitative study of fossils of extinct stylophorans (*Lagynocystis* and *Jaekelocarpus*), widely accepted as deuterostomes, utilized sophisticated technology to image and reconstruct internal anatomical structures thought to be involved in suspension feeding or respiration. While these have been called gill bars by earlier workers. The actual placement of the group as echinoderms or early chordates is a matter still unsettled in the literature but is key to the overall conclusions. The disputed affinity is not mentioned. Nevertheless, the current study considers them to be early echinoderms and presents quantitative measurements of dimensions of these gill bars and compares them with gill bars of extant hemichordates and cephalochordates. Additional datasets generated from "bar-like elements" thought to be respiratory structures in 2 other definitive, but extinct, echinoderm groups (blastoids and rhombiferan) are added for additional comparison.

We appreciate the referee's comments. We strongly agree with the historically contentious positioning of the group; however, the recent discovery of several exceptionally preserved specimens of *Thoralicystis* nov. sp. and *Hanusia* nov. sp. from the Lower Ordovician of Morocco provides unequivocal evidence of a water vascular system preserved within the

proximal part of the appendage and the presence of ambulacral structures (Lefebvre *et al.* 2019). This evidence supports the placement of stylophorans within Echinodermata as they possess at least two of the major synapomorphies thought to define the Phylum: i) an endoskeleton formed of multiple elements composed of single crystals of calcite with a mesh-like microstructure (stereom) and ii) a water vascular system. The affinities of stylophorans are now briefly discussed in the introduction [lines 67–77].

The images of reconstructions of the stylophoran *Lagynocystis* are exquisite (but too small) and add to earlier, similar images (Dominguez *et al.*) of the other stylophoran (*Jaekelocarpus*) used in the dataset.

We appreciate the referee's positive comments on our images. These images were already increased in size as part of our previous revision and unfortunately cannot be enlarged further due to space limitations. However, we have provided all the 3D models on which they are based as supplementary data, so interested readers will be able to look at any parts of the specimens in much more detail than possible from figures alone.

The multidimensional analyses show that dimensions of gill bars in stylophorans clearly and strongly overlap those of the chordate groups but not those of the echinoderm groups. These data support morphological homology of the gill bars of stylophorans and chordates. They could also be convergent structures.

As the referee alludes to, our results reveal statistically significant similarities between the internal bars of stylophorans and the gill bars of extant deuterostomes. This is consistent with some previous studies that described close qualitative similarities between these structures (e.g. Jefferies 1973; Dominguez *et al.* 2002). Together, this strongly supports the morphological homology of gill bars in stylophorans and extant deuterostomes.

There are some notable differences between the internal bars of the stylophorans and the gill bars of extant deuterostomes, in particular the number of bars, which increase in number during ontogeny [lines 233–238] and the absence of secondary bars [lines 226–228]. We interpret these differences as the product of heterochrony [lines 231–233]. See comment below.

Based on this dataset and the assumption that stylophorans are echinoderms the manuscript argues that “the latest common ancestor of echinoderms, hemichordates and chordates was a bilaterally symmetrical worm with pharyngeal openings, with these characters lost in echinoderms” (MS Abstract, lines 23–25). The argument is complex and hard to follow as presented. Several current developmental phenomena/processes (heterochrony and deep homology are invoked to rationalize the inference that gill bars should be present in the last common ancestor or chordates and echinoderms.

We have addressed this in our comments on heterochrony and deep homology below.

I am a reviewer of what appears to be a thoroughly reviewed and revised MS; comments from 4 other reviewers and responses of authors were included in my “review package”. While I've read these comments and responses I do not have the time or expertise to arbitrate the revisions. I will give a few comments on my reaction to the current MS.

The images and datasets are important and valuable contributions. The images in the

supplementary material are also important for context. Perhaps they should be combined with those of the main text (and those in Dominquez et al) and made into a larger manuscript. (I think a more expansive manuscript directed to a more specialized audience would have a greater, longer lasting impact (than the present MS) on thinking about gill bars and how stylophorans may have worked).

We appreciate the referee's comments, but strongly disagree our work would be better suited to a more specialist journal. As highlighted by this and previous reviewers, we present the first rigorous quantitative analysis of gill bar-like structures across a range of extinct and extant deuterostomes, which represents a major step forward in the field (as noted by all five referees). We agree with the referee that an in-depth study of gill bar formation and evolution in deuterostomes is highly desirable and will be of a great impact for all specialised workers in this field; however, this is only part of the contribution made by our manuscript. Importantly, our results strongly suggest that the internal structures preserved in some fossil stylophorans are homologous to the gill bars of extant deuterostomes. This has major implications for understanding the evolution of pharyngeal openings in echinoderms and across deuterostomes. Consequently, we believe our findings will be of great interest to researchers working on deuterostome evolution, comparative biology, evolutionary genetics and palaeontology, and we consider that Proceedings B is the ideal journal with which to reach these diverse biological audiences.

The phylogenetic position of stylophorans is a key assumption and not sufficiently discussed in the MS. So the whole argument is "If stylophorans are echinoderms then" What if they are early chordates? Related is that stylophorans studied here are not the earliest of their clade and the gill bars described here are not reported from earlier group members. That's curious and not really addressed.

See our answer to comment #1 of this review and expanded introduction [lines 67–77].

"Heterochrony" is defined as a change in timing of expression of a trait relative to that in a related lineage. I find no clear explanation of why it is evoked here. Though the term is in the title, it is mentioned in the text (discussion) only 3 times – 2x when numbers of bars are compared between the groups compared whose affinity is unclear. At the very least it should be dropped from the title. I do not find clear evidence for (between lineage) heterochrony in the statements about numbers of gill bars.

The term "heterochrony" is defined as a change in timing of development of a trait relative to that in a related lineage in *Evolution: A Developmental Approach* by Wallace Arthur (2010) and other works. The evolutionary change that we document in stylophoran fossils is the loss (or gain in modern deuterostomes) of secondary bars with respect to primary bars, and the second example is the differing number of gill bars between taxa [lines 242–244]. We acknowledge that it is hard to compare the relative timing of the acquisition of gill bars in the modern deuterostomes analysed (hemichordates and cephalochordate) to the stylophoran fossils by way of another trait for reference.

The concept of 'deep homology' is also brought into the mix but is not explained and there is no explicit explanation (connecting the dots) of how it applies to stylophorans. If deep homology involves what specifically are the authors thoughts on how it applies to their study organisms?

We use the term deep homology when referring to the presence of various gene clusters associated with pharyngeal development, specifically gill pouch development, expressed during larval stages across all deuterostome groups, this includes several echinoderm groups [lines 55–61 and 335–343]; albeit gill pouches are not developed in any extant echinoderms. We do not possess genetic data for blastozoans or any other extinct echinoderms; however, given the gene clusters are conserved in modern echinoderms and across all known deuterostomes, it is very likely these clusters were also present in blastozoans and stylophorans [lines 344–347]. Since the internal bar-like structures in blastozoan (i.e. hydrospires and rhombs) do not show any statistical similarities to the internal bars in the other and they are widely accepted to have functioned primarily in respiration, any morphological similarities are best explained as the result of parallel evolution.

Another comment is that tentaculate and lamellar structures comprised of few to many parallel cylinders each lined with rows of cilia are common throughout many metazoan phyla and are assumed to have arisen independently many times. Motile cilia that move water for respiration or feeding work best when placed on epithelial cylinders extended out into cavities or on cylinders extended from external body surfaces. It is possible the calcified gill bars of stylophorans are an example of convergence without deep homology. The present multidimensional study targets hemichordates and cephalochordates that are favored to compare with the fossil stylophorans, and tosses in a few fossil echinoderms whose bar-like structures are unlikely structural homologs. How these data stack up against other ‘water moving’ (suspension feeding) organs is not considered.

We agree with the referee that superficially similar gill-like structures are also found in other non-deuterostome metazoans and this is now mentioned in the discussion [lines 289–292]. However, there are important morphological similarities between the internal bars of stylophorans and deuterostome gill bars, including the length and width of the individual bars and the spacing between them, which suggests similar sized ciliated cells lined these structures. Moreover (as mentioned in the manuscript), genetic evidence strongly supports the homology of deuterostome pharyngeal structures [lines 334–342]. See comment above.

In my opinion this study lacks the breadth and depth of data to support the discussion. However, the images and quantitative data are excellent contribution to a more specialized journal.

We thank the referee for their helpful comments, which have allowed us to strengthen this study by clarifying several points that would have been unfamiliar to a non-specialist reader. As noted above, we strongly disagree our paper would be better suited to a more specialized journal. The positive comments we have received from the editor and other reviewers also point to the suitability of our work for publication in Proceedings B.